# An Amish founder population reveals rare-population genetic determinants of the human lipidome

May E. Montasser [1✉], Stella Aslibekyan[2,7], Vinodh Srinivasasainagendra[2], Hemant K. Tiwari[2], Amit Patki[2], Minoo Bagheri[2,3], Tobias Kind[4], Dinesh Kumar Barupal[4], Sili Fan[4], James Perry [1], Kathleen A. Ryan [1], Alan R. Shuldiner[5], Donna K. Arnett [6], Amber L. Beitelshees[1], Marguerite Ryan Irvin[2] & Jeffrey R. O'Connell[1]

Identifying the genetic determinants of inter-individual variation in lipid species (lipidome) may provide deeper understanding and additional insight into the mechanistic effect of complex lipidomic pathways in CVD risk and progression beyond simple traditional lipids. Previous studies have been largely population based and thus only powered to discover associations with common genetic variants. Founder populations represent a powerful resource to accelerate discovery of previously unknown biology associated with rare population alleles that have risen to higher frequency due to genetic drift. We performed a genome-wide association scan of 355 lipid species in 650 individuals from the Amish founder population including 127 lipid species not previously tested. To the best of our knowledge, we report for the first time the lipid species associated with two rare-population but Amish-enriched lipid variants: APOB_rs5742904 and APOC3_rs76353203. We also identified novel associations for 3 rare-population Amish-enriched loci with several sphingolipids and with proposed potential functional/causal variant in each locus including GLTPD2_rs536055318, CERS5_rs771033566, and AKNA_rs531892793. We replicated 7 previously known common loci including novel associations with two sterols: androstenediol with UGT locus and estriol with SLC22A8/A24 locus. Our results show the double power of founder populations and detailed lipidome to discover novel trait-associated variants.

[1] Division of Endocrinology, Diabetes and Nutrition and Program for Personalized and Genomic Medicine, Department of Medicine, University of Maryland School of Medicine, Baltimore, MD, USA. [2] Department of Epidemiology, University of Alabama at Birmingham, Birmingham, AL, USA. [3] Department of Cardiovascular Medicine, Vanderbilt University Medical center, Nashville, TN, USA. [4] West Coast Metabolomics Center, Davis, CA, USA. [5] Regeneron Genetics Center, LLC., Tarrytown, NY, USA. [6] Department of Epidemiology, University of Kentucky, Lexington, KY, USA. [7] Present address: 23andMe Inc., Sunnyvale, CA, USA. ✉email: mmontass@som.umaryland.edu

Cardiovascular disease is the leading cause of death worldwide[1]. Besides the well-known role of traditional lipids (total [TC], low-density lipoprotein [LDL], and high-density lipoprotein [HDL] cholesterol and triglycerides [TG]) in CVD risk and progression, molecular lipid species (lipidome) were also found to be an independent contributors[2]. Previous studies identified ceramides as a key player in atherogenesis[3], found variable effects of phospholipids and TG species on CVD based on the degree of saturation[4–7], and was able to improve risk prediction by adding lipid species[6,8]. Identifying the genetic determinants of inter-individual variation in lipidome may provide deeper understanding beyond traditional lipids, and may lead to additional insight into the mechanistic effect of lipid variants and their role in CVD risk and progression[2]. Previous studies tested lipidome genetic determinants either as a small part of large metabolite studies or in a small number of candidate lipid species (full list of studies available in Hagenbeek[9]), with the exception of a published study that performed a focused lipidome genome-wide association scan (GWAS) for 141 lipid species in 2181 Finnish individuals. Here, we performed a GWAS in 650 individuals from the Old Order Amish (OOA) founder population using an expanded number of 355 lipid species from 14 classes that included 127 not previously tested for genetic association. Founder populations can facilitate the identification of previously unknown disease associations with variants that are enriched to a higher frequency through genetic drift. Multiple examples have been recently reported of highly enriched variants with large effect sizes associated with complex diseases and traits in homogenous populations in Iceland[10], Sardinia[11], Greenland[12], Samoa[13] and OOA[14–21]. While such drifted variants are often rare or absent in the general population, their associations can inform biological mechanisms and therapeutic targets relevant to all humans. The population-based Genetics of Lipid Lowering Drugs and Diet Network (GOLDN) study[22–25] was used for replication and fine mapping, and publicly available association results databases from several large biobanks were used to look up the top results. We identified five rare-population but Amish-enriched loci, three of which are novel, and replicated 7 previously known common loci including two loci with novel trait associations. These results demonstrate the power of detailed lipidome profiling in a founder population to identify novel rare variants enriched through genetic drift to accelerate lipid loci discovery and substantially advance our understanding of the genetic role in lipid biology.

## Results

**Additive and dominant heritability.** The narrow sense heritability, defined as the ratio of additive variance to phenotypic variance, was estimated for each lipid species and traditional lipid using a mixed model with pedigree kinship covariance matrix. We also tested if dominance variance contributes to lipidome genetic architecture by comparing the additive model to a model that included a dominance and additive effect using a likelihood ratio test. No lipid species or traditional lipid showed significant dominant variance after Bonferroni correction, indicating that the lipidomic genetic architecture is primarily additive.

The full list of heritability estimates of the 355 lipid species measured in 650 individuals from the Old Order Amish (OOA) founder population (Supplementary Data 1) with and without adjustment for 4 Amish-enriched large effect lipid variants (*APOB*_rs5742904[15], *APOC3*_rs76353203[14], *B4GALT1*_rs5515646 83[21], *TIMD4*_rs898956003[20]) (4 variants) is provided in Supplementary Data 2. Figure 1a shows that the heritabilities range between 0 and 0.7, with significant attenuation when adjusting for the 4 variants as they account for a significant proportion of the

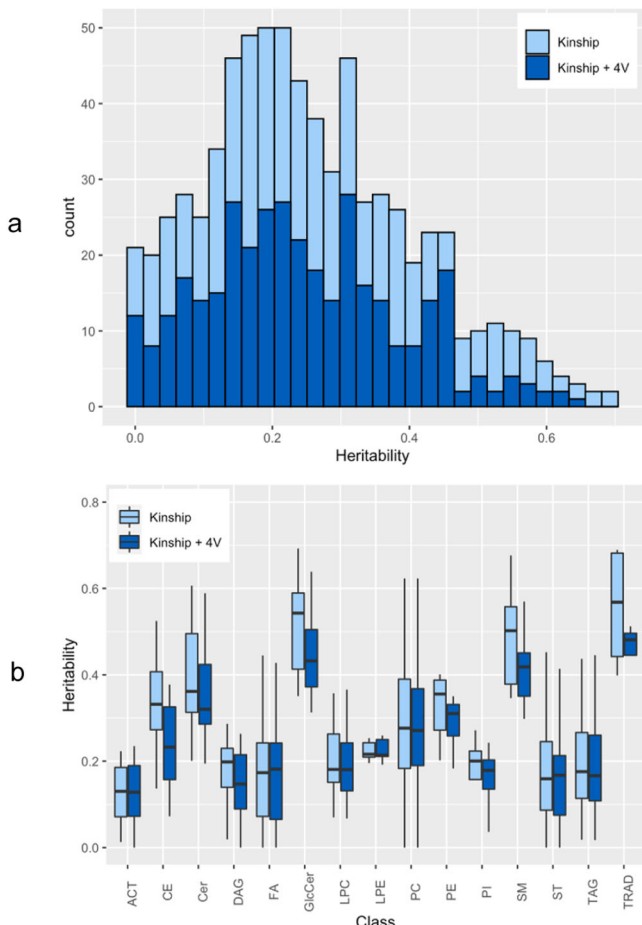

**Fig. 1 Heritability of the lipid species. a** Histogram showing the heritability distribution for all lipid species. **b** Box plot for the heritability by class. Heritabilities presented as unadjusted and adjusted for 4 Amish-enriched lipid variants (*APOB*_rs5742904[15], *APOC3*_rs76353203[14], *B4GALT1*_rs551564683[21], *TIMD4*_rs898956003[20]). The upper, center, and lower line of the boxplot indicates third quartile (Q3), median, first quartile (Q1), respectively. The upper and lower whisker of the boxplot indicates Q3 + 1.5 * interquartile range (IQR) and Q1 − 1.5 * IQR. Outliers are suppressed from the plot for readability. ACT Acylcarnitine, CE Cholesteryl ester, Cer Ceramide, DAG Diglycerides, FA Fatty acids, GlcCer Glycosphingolipids, LPC Lysophosphatidylcholines, LPE Lysophosphatidylethanolamines, PC Phosphatidylcholines, PE Phosphatidylethanolamine, PI Phosphatidylinositol, SM Sphingomyelin, TAG Triglycerides, ST Sterols, TRAD Traditional lipids.

phenotypic variance. The (near-) zero estimates reflect potential lack of genetic contribution to the lipid species. The histogram suggests a bi-modal distribution with second mode near 0.55 driven mainly by sphingolipids including ceramides (Cer), sphingomyelins (SM) and glycosphingolipids (GlcCer). Figure 1b shows heritability estimates for each lipidome class with and without 4 variants adjustment. Each class has non-zero median heritability, and most classes show considerable variability. The highest heritability was reported for GlcCer (0.35 – 0.69) while acylcarnitines (ACT) was the lowest (0.01 - 0.22). Consistent with previous reports[26–28], we found sphingolipids to have higher heritability than glycerolipids. The contribution of the 4 variants varied across class. The classes with the biggest difference were cholesteryl ester (CE), Cer, GlcCer, SM and phosphatidylethanolamine (PE) primarily driven by the LDL-increasing *APOB*_rs5742904 variant. Many classes showed little change in median or variation, including triglycerides (TAG), where the overall impact of *APOC3*_rs76353203 null variant on heritability

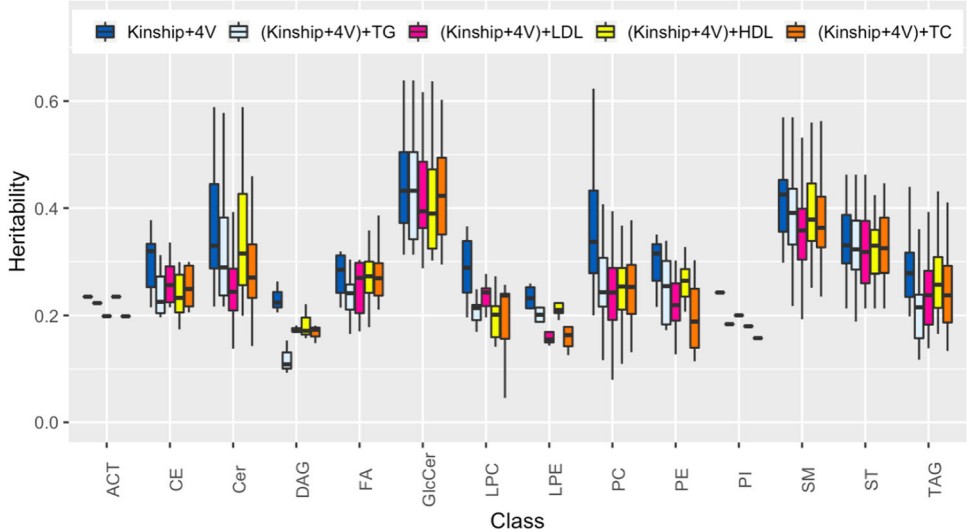

**Fig. 2 Contribution of previously identified lipid GWAS variants to lpidomics variance.** Heritability estimates using adjusted kinship separately (Adjusted for 4 Amish-enriched lipid variants (*APOB*_rs5742904[15], *APOC3*_rs76353203[14], *B4GALT1*_rs551564683[21], *TIMD4*_rs898956003[20]) and then jointly with a SNP GRM for each traditional lipid. Analysis restricted to 194 lipid species with baseline significant heritability (Kinship + 4 V heritability *p*-value < 0.01). The upper, center, and lower line of the boxplot indicates third quartile (Q3), median, first quartile (Q1), respectively. The upper and lower whisker of the boxplot indicates Q3 + 1.5 * interquartile range (IQR) and Q1 − 1.5 * IQR. Outliers are suppressed from the plot for readability. ACT Acylcarnitine, CE Cholesteryl ester, Cer Ceramide, DAG Diglycerides, FA Fatty acids, GlcCer Glycosphingolipids, LPC Lysophosphatidylcholines, LPE Lysophosphatidylethanolamines, PC Phosphatidylcholines, PE Phosphatidylethanolamine, PI Phosphatidylinositol, SM Sphingomyelin, TAG Triglycerides, ST Sterols.

was small. The difference between the two variants can be explained by their different impact on traditional lipids: *APOC3*_rs76353203 accounts for ~3% of the TG variance (h2 = 0.398 vs 0.368 adjusted) whereas *APOB*_rs5742904 accounts for ~18% of LDL-C variance (h2 = 0.68 vs 0.5 adjusted).

**Genetic contribution of traditional lipids to lipidome**. To estimate the genetic contribution of previously identified traditional lipid GWAS variants to lipidome variance, a SNP genetic relationship matrix (SNP GRM) was constructed using the variants and included with the kinship matrix in the mixed model for joint variance estimation. The impact of SNP GRM on the heritability estimates in the joint model compared to baseline heritability with no SNP GRM is shown in Fig. 2. In general, known lipid GWAS variants had a small contribution to the genetic variation which vary between classes and by lipid SNP GRM within class. The greater the contribution of the SNP GRM to lipidome class variance, the lower the residual heritability estimates. For example, between lipid SNP GRM variability is small within sterols but significant within Lysophosphatidylcholines (LPC) and within Lysophosphatidylethanolamines (LPE). HDL and TG associated variants almost explain no variance of SM while LDL and TC associated variants explain over 15% of SM, which agrees with the role of SM and cholesterol in the structure of plasma membranes. TG associated variants explain a larger proportion of TAG than other lipid associated variants, as expected.

**Genetic and phenotype correlation**. Pairwise genetic and phenotypic correlation for 355 lipid species and 4 traditional lipids combined are shown in Supplementary Data 3 and Supplementary Fig. 1 (heatmap). In general, genetic and phenotypic correlation were lower between classes than within classes. SMs, TAGs, and diglycerides (DAGs) exhibited the strongest within class correlation, and as expected the strongest between class correlation was found for TAGs with DAGs. While TAGs exhibited the

strongest within class correlation, we found that the correlation between TAG pairs where both species have > = 54 carbons and > = 4 double bonds were significantly stronger and less variable (*p* < 2.2E-16) than correlation between pairs where one or both species have <54 carbons and <4 double bonds (Supplementary Fig. 2). The phenotypic correlations have both smaller median values and less variance than genetic correlations even with larger number of pairs due to phenotypic correlations having greater precision since there is no maximum likelihood estimation required.

The correlation with traditional lipids were also generally limited, with the exception of TAGs and DAGs that had the strongest positive genetic correlations with traditional TG, and the strongest negative correlation with HDL. These results are in line with a previous finding[26] and explain the limited contribution of traditional lipid genetics to the lipid species (Fig. 1b). This limited overlap highlights the value that lipid species would contribute to understanding CVD risk factors beyond traditional lipids[8].

**Lipidome contribution to traditional lipids**. Understanding the relative contribution of each lipid species to traditional lipid and the interplay between components will help us to gain insight into their architecture. The estimated proportion of each traditional lipid variance explained by kinship and each lipidome class are shown in Supplementary Fig. 3. The lipidome class was included in the mixed model by constructing covariance matrix between the species in the class (see Methods). All classes explained a significant proportion of lipid variation with different magnitudes (Supplementary Data 4). For example, while PC was the most statistically significant class for HDL and LDL, it was the 2nd for TC and the 3rd for TG. Not surprisingly, the most significant class across all lipids is TAG with TG. The least significant class on average was acylcarnitines (ACT).

We also performed a sequential analysis to determine which lipidome classes jointly with the kinship explain the greatest amount of variance of each traditional lipid. Supplementary

Data 5 shows the decomposition with the class, remaining unexplained heritability, and residual error variance estimates. HDL maxed out at 3 classes, LDL and TC at 5 classes and TG at 4 classes. Compared to the single class model in Supplementary Data 4 the magnitude and precision of the estimates in the multi-class models may differ due to potential correlation between classes. The heritability estimates in the multi-class are reduced to less than 0.16 as more of the additive variance is accounted for by additional lipidome classes. The decomposition differs by lipid. TG is primarily composed of TAG (34%) with DAG and PC, accounting for ~ 5%, while the 3 other lipids have at least two classes with high proportions. LDL has the lowest residual variance at 20% indicating the phenotypic architecture of LDL may be more influenced by lipidome than other lipids. Overall, the variance component analyses show that lipidome classes contribute a significant portion of the variance of traditional lipids but there remains 10-15% heritability unexplained by lipidome, which again indicate the differences in genetic architecture.

**GWAS results**. We performed a GWAS for 355 lipid species with ~8 million genetic variants in 639 Amish individuals with both phenotype and genotype information. We identified 12 significantly associated signals ($p < 4.5E-10$, using 5E-08/110, based on the first 110 principal components explaining > 95% of the variance in the 355 lipid species), five were Amish-enriched rare-population variants, three of which have not been previously reported, and seven were common variants that were previously associated with lipid species (Table 1, Figs. 3 and 4).

The genetic architecture of the Amish is characterized by long runs of homozygosity as a result of founder effects[29], so the Amish-enriched associated loci are usually long haplotypes with many variants in strong LD, making it difficult to statistically separate variants to identify the potential causal variant. All results with $p < 5E-08$ are listed in Supplementary Data 6.

*Rare-general population but Amish-enriched loci.* The most interesting finding among the five Amish-enriched loci is a rare population missense variant rs536055318 (A263T) (MAF = 0.07 vs 0.001 in the general European population) in an active transcription start site (aTSS) within the promoter region of the glycolipid transfer protein domain containing 2 (*GLTPD2*) gene on chromosome 17 that was strongly associated with lower level of SM(d40:0) ($p = 1.1E-12$) and suggestively associated with SM(d36:0, d38:0). To the best of our knowledge, these 3 SMs have never been previously interrogated for genetic association. Another independent African enriched variant (rs73339979) downstream of *GLTPD2* was previously associated with lower total and LDL cholesterol[30]. Also, a Finnish-enriched *GLTPD2* intronic variant (rs79202680) was recently associated with lower level of several SMs and reduced atherosclerosis[26].

The second interesting Amish-enriched but rare-population finding (MAF = 0.04 vs 0.01) was on a 5 Mb long haplotype on the short arm of chromosome 12 that was significantly and suggestively associated with lower levels of SM(d32:2) and SM(d30:1), respectively. Other independent variants in this region were previously associated with alanine, 1,5-anhydroglucitol (1,5-AG), and creatine, but not with any lipid species. One of the top variants is a splice donor missense variant (rs771033566, Val344Leu, $p = 2.2E-14$) in the ceramide synthase 5 (*CERS5*) gene and classified as disease-causing by the Mutation Taster software[31]. Another common coding variant in this gene was previously associated with increased systolic/diastolic blood pressure and hypertension[32,33]. The sphingolipid metabolic pathway has been previously linked to blood pressure regulation and

**Table 1 Genomic loci significantly associated with lipid species.**

| SNP | Position | A1/A2 | A2_Freq_Amish | A2_Freq_Eur | Top associated trait | Effect[a] | P-value | Gene | Type | GOLDN effect[a] | GOLDN P-value |
|---|---|---|---|---|---|---|---|---|---|---|---|
| **Rare-population Amish-enriched loc** | | | | | | | | | | | |
| rs5742904 | 2:21006288 | C/T | 0.0634 | 0.0004 | **SM(d34:1)** | 1.22 | 7.89E-25 | APOB | missense | - | - |
| **rs7863920** | **9:115987317** | **G/A** | **0.0433** | **0.0169** | **GlcCer(d42:2)** | **-1.14** | **6.22E-18** | **LINC00474** | **intergenic** | - | - |
| rs76353203 | 11:116830637 | C/T | 0.0219 | 0.0008 | **PE(36:2)** | -1.45 | 6.29E-13 | APOC3 | nonsense | - | - |
| **rs147698408** | **12:50001009** | **T/C** | **0.0447** | **0.0159** | **SM(d32:2)** | **-1.06** | **1.79E-14** | **RACGAP1** | **intronic** | - | - |
| **rs536055318** | **17:4790207** | **G/A** | **0.0762** | **0.0014** | **SM(d40:0)** | **-0.76** | **1.14E-12** | **GLTPD2** | **missense** | - | - |
| **Common known loci** | | | | | | | | | | | |
| rs887829 | 2:233759924 | C/T | 0.4194 | 0.3272 | androstenediol | 0.44 | 1.23E-15 | UGT1A3,10 | intronic | -0.02 | 7.25E-01 |
| rs3778167 | 6:11033235 | T/C | 0.2771 | 0.4233 | PC(42:5) | -0.40 | 3.65E-10 | ELOVL2 | intronic | -0.24 | 3.78E-06 |
| rs174578 | 11:61838027 | T/A | 0.2515 | 0.3552 | PC(37:4) | -0.78 | 6.33E-36 | FADS2 | intronic | -0.43 | 2.32E-14 |
| rs184061227 | 11:63073643 | A/G | 0.0555 | 0.0714 | estriol | 0.98 | 1.05E-15 | SLC22A24 | intergenic | -0.22 | 2.70E-02 |
| rs10468017 | 15:58386313 | C/T | 0.2621 | 0.2869 | PE(36:4) | 0.48 | 7.32E-15 | LIPC | intergenic | 0.28 | 2.76E-07 |
| rs321940 | 20:12979237 | A/G | 0.5239 | 0.6014 | Cer(d43:1) | -0.69 | 3.37E-32 | LOC101929486 | intergenic | -0.45 | 2.95E-19 |
| rs360525 | 20:130020644 | G/A | 0.0728 | 0.1240 | Cer(d43:1) | 0.86 | 5.34E-15 | SPTLC3 | intronic | 0.19 | 7.58E-03 |

A1: Non-coded allele
A2: Effect allele
A2_freq: Effect allele frequency
Position: Variant position according to hg38
aEffect size in standard deviation units
In bold: Novel locus, or novel trait in a known locus
Cer: Ceramide
GlcCer: Glycosphingolipids
PC: Phosphatidylcholines
PE: Phosphatidylethanolamine
SM: Sphingomyelin

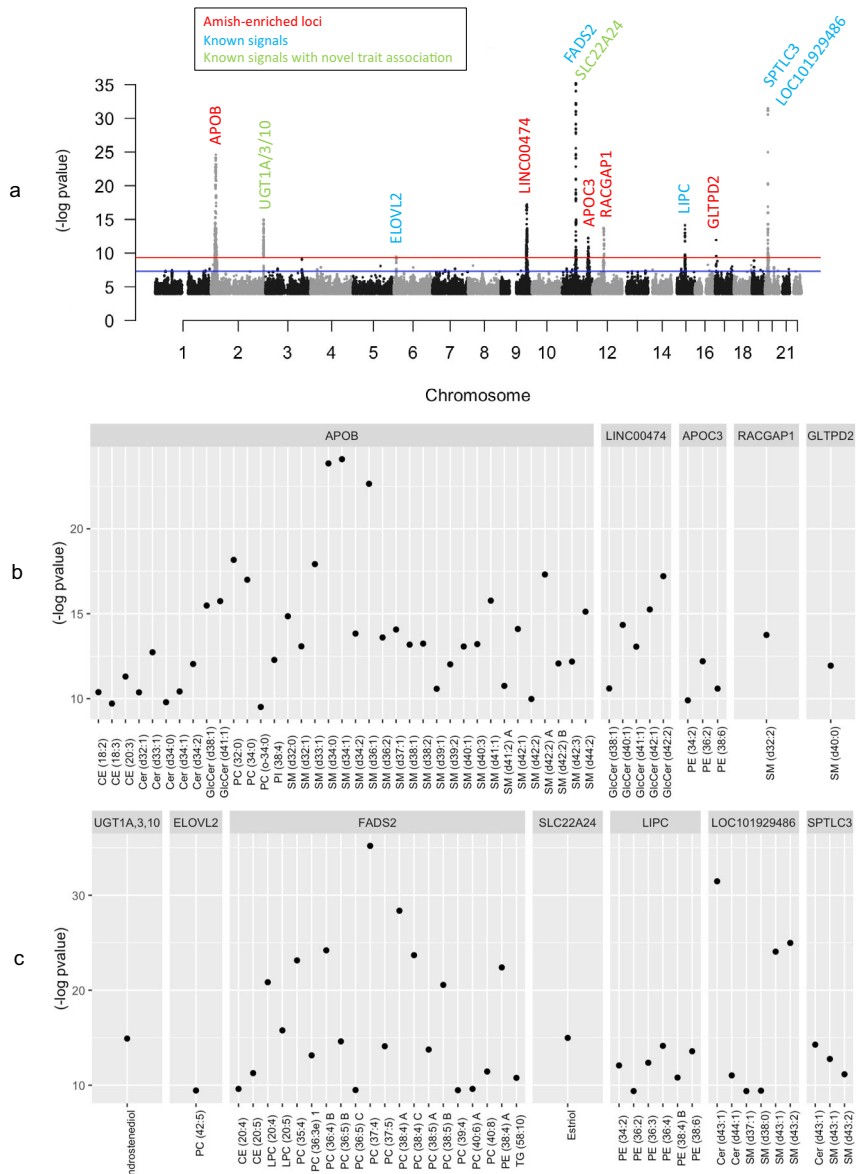

**Fig. 3 Lipidomic association results. a** Manhattan plot for the association results of all 355 lipid species in 639 Amish subjects. Amish-enriched loci denoted in red, previously known signals denoted in blue and previously known signals with novel trait association and/or novel variant in known loci in green. Blue line marks a genome-wide suggestive threshold (5.0E-08) and red line marks a genome-wide significant threshold (4.5E-10). **b** GWAS results for all significantly associated lipid species in Amish-enriched loci. **c** GWAS results for all significantly associated lipid species in previously known loci. All *p*-values based on t-test using additive genetic model. CE Cholesteryl ester, Cer Ceramide, GlcCer Glycosphingolipids, PC Phosphatidylcholines, PE Phosphatidylethanolamine, SM Sphingomyelin.

response to thiazide diuretics[34–36], suggesting that CERS5 may affect blood pressure level through alteration of sphingolipids.

Another Amish-enriched 8 Mb long haplotype (MAF = 0.04 vs 0.01 for the top variant) on the long arm of chromosome 9 was strongly associated with lower levels of all tested glucosylceramide species (GlcCer(d38:1), (d40:1), (d41:1), (d42:1), (d42:2)) except the one with the shortest acyl chain (GlcCer(d34:1)), which reflects the strong phenotypic correlation between the first 5 ($r = 0.6–1.0$) compared to their much lower correlation with GlcCer(d34:1) ($r < 0.2$). Other independent variants in this region were previously associated with total cholesterol[37], urate, p-acetamidophenylglucuronide, and LPC(28:0)A[9]. Based on the pattern of the association results (Fig. 4b), we expect the functional variant to be one of the top 27 variants with *p*-values < 8.5E-16 and $r^2 > 0.75$ with the top variant

(Supplementary Data 7). These 27 variants are located within 9 genes (*LINC00474, ATP6V1G1, C9orf91, LOC100505478, DFNB31, LOC101928775, DEC1, AKNA, and COL27A1*), none of which are obvious candidate genes. Formal fine mapping analysis using PAINTOR[38] with different parameters and functional information consistently identified the top associated variant (rs7863920, *p* = 6.2E-18) to have the highest posterior probability of causality at 0.87. Functional annotation highlighted one intronic variant (rs531892793, *p* = 3.9E-17) as a strong potentially functional variant (Supplementary Table 1). This variant is highly enriched in the Amish (MAF = 0.04 vs 0.0001) and located in a promoter flanking region in the AT-hook transcription factor (*AKNA*) gene; it has the top ENCODE DNase score of 1000 indicating very strong evidence of a DNase I hypersensitivity site[39], an eigenPC score of 3.5 (top 1%)

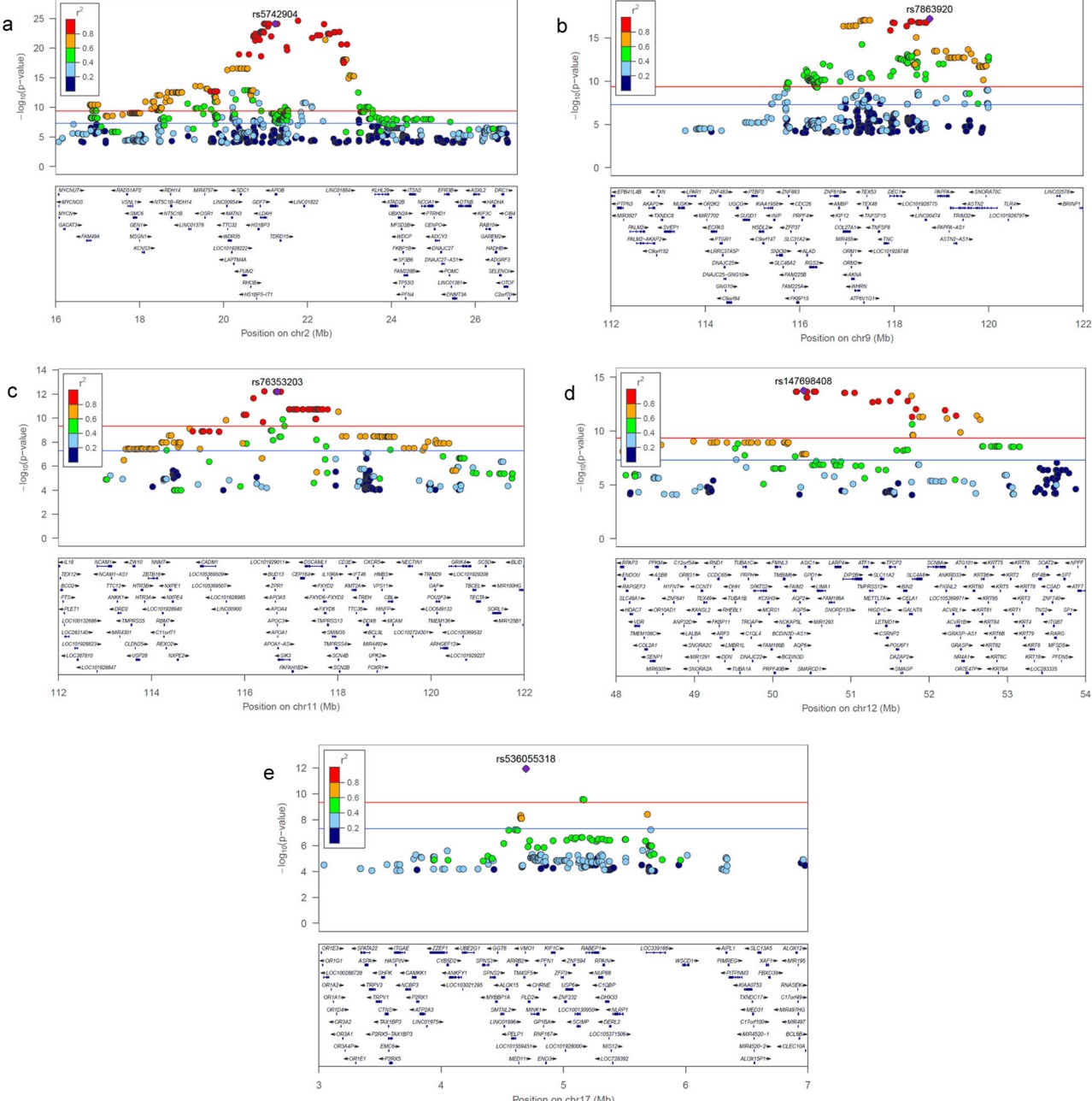

**Fig. 4 Rare-population but Amish-enriched loci.** Locus zoom for 5 loci in **a**, chromosome 2 with SM(d34:1) **b**, chromosome 9 with GlcCer(d42:2) **c**, chromosome 11 with PE(36:2) **d**, chromosome 12 with SM(d32:2) and **e**, chromosome 17 with SM(d40:0). All p-values based on *t*-test using additive genetic model. GlcCer Glycosphingolipids, PE Phosphatidylethanolamine, SM Sphingomyelin.

indicating a strong functional prediction based on conservation and allele frequency[40], and is predicted to affect transcriptional factor binding with a 2a RegulomeDB classification[41]. The variant is located in a weak transcription site in the islet and skeletal muscle, in a genic enhancer region in liver tissue, and in an active enhancer region in adipose tissue[42].

We also have two well-established Amish-enriched variants that we previously reported their strong association with traditional lipids, but have never been interrogated for association with lipid species. The first is the missense variant R19X (rs76353203) in the *APOC3* gene (MAF = 0.02 vs 0.0008) that we first reported its association with lower TG, higher HDL, and cardioprotection[14]. In this analysis, we report significant association of this variant with lower levels of 3 phosphatidylethanolamines (PE(36:2), (38:6),

(34:2)) and suggestive association with lower level of another PE, one di- and three triglyceride species. The second is the well-established Amish-enriched familial hypercholesterolemia (FH) causing variant R3527Q (rs5742904) in the *APOB* gene (MAF = 0.06 vs 0.0004) that was previously linked to LDL and TC by our group and others[15,43]. As expected, this variant was significantly associated with increased levels of several cholesterol esters, sphingolipids and phospholipids while there was no association with acylcarnitine, fatty acids, sterols, and glycerolipids.

*Common known loci.* We also replicated 7 previously well-known lipid signals including *UGT1A/3/10* genes on chromosome 2, *ELOVL2* gene on chromosome 6, *SLC22A8/A24* genes and *FADS* genes on chromosome 11, *LIPC* region on

chromosome 15, and 2 independent signals in the *SPTLC3* region on chromosome 20.

A ~500 kb haplotype at the end of chromosome 2 in a region with a cluster of several uridine diphosphate glucuronosyltransferase (*UGT*) genes was strongly associated with higher levels of androstenediol. UGT transforms small lipophilic molecules, such as steroids, bilirubin, hormones, and drugs, into water-soluble, excretable metabolites. Our top variant (rs887829) was previously associated with lower LDL[37] and higher bilirubin[44], however the association with androstenediol is novel.

We also identified a novel strong association for a 300 kb haplotype on chromosome 11 with increased estriol level. The top associated variant (rs184061227, $p = 1.0E-15$) located in a previously known region encompassing *SLC22A8/A24*, which are expressed only in kidney. This region was previously associated with etiocholanolone glucuronide (ETIO-G), which is an endogenous, naturally occurring metabolite of testosterone[45].

The nearby *FADS* region on chromosome 11 was the most significant ($p = 6.3E-36$) and associated with 29 different lipid species including many phosphatidylcholines and cholesterol esters consistent with previous reports[9,26].

We also replicated 2 additional known common loci. The first within the fatty acid elongase 2 (*ELOVL2*) gene on chromosome 6 was associated with PC(42:5) consistent with the previous association of the same variant with DHA_DPAN3 (docosahexaenoic acid, or DHA(22:5)), and (docosapentaenoic acid (DPA) (22:6) omega3)[9]. The second was the well-known lipid loci lipase C, hepatic type (*LIPC*) gene region on chromosome 15, associated with several phosphatidylethanolamines (PEs), similar to previous reports[9,26].

Finally, we replicated two overlapping but independent signals on chromosome 20 within the serine palmitoyltransferase long chain base subunit 3 (*SPTLC3*) gene that encodes a subunit of the SPTLC complex which catalyzes the rate-limiting step in sphingolipid biosynthesis. Consistent with previous reports[9,26], both signals were associated with several ceramides and sphingomyelins, the first signal is very common (MAF = 0.47) and associated with decreased levels, while the second was less common (MAF = 0.07) and associated with increased levels of lipid species.

## Replication/fine mapping in GOLDN

Replicating Amish-enriched rare population loci can be a challenge due to the rarity or absence of variants in outbred populations. However, outbred populations can provide evidence of exclusion even when only a few copies are present as the LD between the causal and non-causal variants that confounds the Amish signal is absent or reduced. If the causal variant is present, it will generally show strong validation with few copies depending on effect size, but non-causal variants will not replicate even if expected replication power is extremely high. The familial hypercholesterolemia causing *APOB* variant rs5742904_R3527Q[15,43] which is enriched in the Amish provides an extreme example. The variant increases LDL by ~50 mg/dl and has a p-value=7.8E-25 in our 639 Amish, and through LD generates genome-wide significant signals at 441 surrounding variants in a 10MB region. Those associations disappear when the Amish LD is accounted for in a conditional analysis with rs5742904 (Supplementary Data 7). Fifty out of the 441 variants were absent in GOLDN, including rs5742904. The remaining 391 variants (MAF: 0.001-0.48) were non-significant in GOLDN, providing confidence they are non-causal. We also performed the same analysis with the *APOC3* TG lowering causal variant rs76353203_R19X on chromosome 11, which was also absent in GOLDN, and all R19X LD-driven significant Amish variants were not significant in GOLDN. These two examples

support applying this approach to the other 3 Amish-enriched loci that we identified, where the causal variant is unknown and most likely not in GOLDN. Power calculations using the observed Amish effect size (or half to adjust for winner's curse) can quantify exclusion thresholds for given variants found in outbred samples. The fine-mapping approach provides a reduced set of potential variants for future follow-up.

For common variants, look up in GOLDN provides direct replication. The basic demographic and clinical characteristics of the GOLDN replication cohort are presented in Supplementary Data 1. All GOLDN association results for our top results are listed in Supplementary Data 6. We had two novel trait associations for androstenediol and estriol. These two sterol lipids did not replicate, however, these two variants had p-values of 1.9E-04 and 2.9E-04 with PC(36:4)A, and PC(38:4), respectively in GOLDN. The other five known significant common loci in the Amish had p-values between 7.5E-03 and 1.4E-35 in GOLDN.

Full data for the 5 Amish-enriched loci with the GOLDN results are shown in Supplementary Data 7. The table and the locus zoom plots in Fig. 4 show that each of these 5 loci is a long haplotype ranging from 4 Mb to 10 Mb.

The *GLTPD2* locus on chromosome 17 has 13 variants with $P < 5.0E-08$ (3 significant), 12 of which were not significant in GOLDN, despite all being common (MAF > 0.07) and some with more carriers than Amish (MAC 110-389), while the missense top variant rs536055318 was absent. More importantly this top variant was the only one out of the 13 variants that was suggestively associated with lower level of TG in UKBB ($p = 6.9E-08$), further supporting our hypothesis that it is the most probable functional variant in this region, pending experimental validation.

The chromosome 12 locus haplotype extends ~5 Mb with 38 significant variants, 15 of which have similar $p$-values ~E-14 due to the strong LD. The top variant was among 30 variants that did not replicate in GOLDN and hence can be excluded as potential functional/causal variants (in particular 18 variants with MAF > 0.015 and power between 0.79 and 0.99 for significant replication). The splice donor missense variant (rs771033566, $p = 2.2E-14$, Val344Leu) in the ceramide synthase 5 (*CERS5*) gene which is our best candidate for causal variant is absent in GOLDN.

The chromosome 9 locus is an 8 Mb long haplotype with 202 significant variants, 27 of which were prioritized based on p-value and LD with the top variant. Only 4 of these 27 variants were present in GOLDN and none was significant including the top variant. Our best candidate causal variant *AKNA*_rs531892793 based on functional annotation was absent in GOLDN.

## Suggestive associations

The GWAS yielded 246 suggestive associations ($4.5E-10 < p < 5E-08$) within 31 loci, 30 of which were previously reported. Among the top suggestive results, we identified an association between ACT(10.0) ($p = 3.5E-08$) and common variants in the *ACADM* gene which encodes the medium-chain specific acyl-Coenzyme A dehydrogenase that plays a role in the fatty acid beta-oxidation pathway and was previously associated with several carnitines (Supplementary Data 6). We also identified an association ($p = 1.3E-08$) between GlcCer(d40:1) and common variants in the ATPase phospholipid transporting 10D (*ATP10D*) gene. Another independent signal in *ATP10D* was previously associated with several glycosphingolipids[46]. These 2 signals were also replicated in GOLDN ($p = 8.4E-07$ and $p = 2.1E-05$, respectively). These results provide added confidence that other signals in our suggestive interval may be true associations but require larger sample size to achieve significance. The only locus that may be considered novel, if replicated, was an association between Cer(d42:2)B and a rare variant (rs79384120, MAF = 0.018,

$p = 8.3E\text{-}09$) in the synuclein alpha interacting protein (*SNCAIP*) gene on chromosome 5, that is linked with Parkinson's disease[47,48], but no known link in this locus to any lipid trait. Ceramides play a role in the physiology and pathophysiology of the central nervous system[49], this role may be genetically determined, at least partially, by *SNCAIP*. This variant did not replicate in GOLDN, indicating that it is either a false positive or the functional variant is another linked variant.

**Lookup of previously identified loci in our results**. While there are many published GWAS for metabolomics, Tabassum[26] is the only published large GWAS that focused on lipidomics and has the most overlap with our study. Thus, we report our results for their top associated variant in Supplementary Data 8. In their Supp Data 2, Tabassum et al.[26] reported 3754 lipidomic-variant pair associations with $p < 5.0 \times 10^{-8}$ comprising 820 variants and 80 lipid species. For 702 variants present in our data, the top associated traits had p-values ranging from 0.051 to 6.3E-36, of which 219 (31%) met the Bonferroni replication significance threshold (0.05/702 = 7.1E-05). We also report our association results for the same trait and variant for 1476 trait-variant pairs, of which 359 (24%) have a *p*-value less than the Bonferroni replication threshold of 3.3E-05, and 1082 (73%) with $p < 0.05$. These Bonferroni thresholds are conservative as they do not account for the correlation between lipid species and the LD between variants. This high level of consistency with previously reported loci highlights the quality of our data and confirms the generalizability of findings in a founder population to outbred populations.

**Association testing using lipidome compared to traditional lipids**. To assess the power of the lipidome to identify genetic signals compared to traditional lipids we tested the association of 226 known lipid associated variants available in Amish with both lipidome and traditional lipids using the same 639 Amish subjects. As previously reported[26], the lipidome showed higher power in identifying the association signals compared to traditional lipids where only *APOB* had stronger association with LDL using the same sample size (Supplementary Fig. 4). Similarly, when we tested the association between 1,602 variants with $p < 5.0\ E\text{-}08$ in any lipid species with traditional lipids in the same sample size, we found strong signals only for *APOB* and *APOC3* (Supplementary Data 9) and only *APOB* had stronger association with LDL.

## Discussion

Here we report the GWAS results for 355 lipid species, the largest number tested in a single study to date.

We identified three novel rare-population variants that are enriched in the Amish on chromosomes 9,12 and 17 that have not been previously associated with any lipid species or traditional lipids. Leveraging results from the GOLDN study we were able to finemap large numbers of variants present on long Amish-enriched haplotypes to identify a potentially functional variant in a biologically plausible gene for each of the three loci.

The first is a missense variant in the promoter of the *GLTPD2* gene that is mainly expressed in liver and kidney and plays a role in the intermembrane transfer of glycolipids but not neutral or phospholipids[50], consistent with its association only with SM(d40:0) in this study. While two independent studies previously pointed to this gene[26,30], neither identified variant had an obvious functional mechanism. The position of this rare missense variant rs536055318 (A263T) in an aTSS within the promoter region of *GLTPD2* can alter its expression leading to lower levels of SM and reduced atherosclerosis[26]. Moreover, rs536055318 was

recently associated with lower levels of TG in UKBB with a suggestive *p*-value of 6.9E-08. This finding is consistent with previously observed changes in cellular lipid metabolism as a result of up and down regulating GLT protein[51]. Several SM species were previously associated with CVD[4,52,53]. Collectively, these findings suggest *GLTPD2* as a potential therapeutic target for CVD protection. Future Mendelian Randomization studies may help to disentangle the direction of causality. This strong association ($p = 1.1E\text{-}12$) with a lower level of SM(d40:0) was identified using only 650 Amish subjects, while it required 461,140 UKBB subjects to find a suggestive association with TG.

The second is a potentially disease-causing splice donor missense variant (rs771033566, Val344Leu) in the *CERS5* gene, associated with lower SM(32:2). Another common coding variant in this gene was previously associated with increased systolic/diastolic blood pressure and hypertension[32,33]. The sphingolipid metabolic pathway was previously linked to blood pressure regulation and response to thiazide diuretics[34–36], suggesting that *CERS5* may affect blood pressure level and drug response through alteration of sphingolipids, which may have personalized medicine implications. *CERS5* is one of the six members of the ceramide synthase gene family which plays a major role in the sphingolipid metabolic salvage pathway[2], and while many genetic variants in *CERS4* have been previously associated with several SMs[2], this is the first association of a SM species with a *CERS5* genetic variant.

The third is an intronic variant (rs531892793) that was associated with lower levels of five glucosylceramide species with acyl chains of 38 or more carbons, but not with the species with 34 carbons. This result is consistent with a recent study that found significantly increased serum levels of only glucosylceramide species with acyl chains of 38 or more carbons among CAD cases compared to controls[8], but not with 2 shorter carbon species. This variant has very strong regulatory function prediction and is located in the widely expressed *AKNA* gene that encodes AT-hook transcription factor. This transcription factor is essential for normal development and immune function, as indicated by the gene name that means 'mother' in Inuit and Mayan language[54]. *AKNA* knock out mice were weak, short lived and suffered from systemic inflamation[55]. Other common variants in *AKNA* were previously associated with TC, HDL, ApoA1, ALT, AST, and testosterone[30,56–58]. Collectively these data support our hypothesis that *AKNA*_rs531892793 is the best potential functional gene and variant in this locus, however more work is needed to confirm this result as we cannot rule out the possibility that some other variant that may failed genotyping or imputation is the real causal variant driving this association signal.

Two well-known rare-population lipid variants that are Amish-enriched and previously reported by our group are the FH variant *APOB*_R3527Q and the cardioprotective *APOC3*_R19X. Given the rarity of these variants in the general population they have never been interrogated for association with lipid species. While this is, to the best of our knowledge, the first report for the associations of these variants with lipidomics as detailed herein, these associations are not unexpected based on the structure and function of associated traditional lipids. The association of the missense rare population variant *APOC3*_R19X with lower TG, higher HDL, and cardioprotection[14] was first reported by us and was later replicated in other studies[59–62] and led to the development of APOC3 antisense molecules that are currently in phase III clinical trials for the treatment of hypertriglyceridemia[63,64]. Similarly, the three novel variants reported here may lead to novel treatment and/or personalized medicine once there is a large enough general population study for replication and functional study to prove causation. Replicating the association of these three novel variants would require larger sample sizes with

similar lipid species measured and whole genome sequence data, which currently does not exist but may soon be available through large consortia like TOPMed[65].

We also replicated seven previously well-known lipid signals including *UGT, ELOVL2, SLC22A8/A24, FADS, LIPC,* and two independent signals in the *SPTLC3* gene, among these seven, we have two cases of novel trait associations in *UGT* and *SLC22A8/A24*. First, in addition to previous associations of *UGT*_rs887829 with lower LDL[37] and higher bilirubin[44], we also found an association with higher androstenediol. This pleiotropic effect may explain the inverse association of bilirubin with LDL[66] and CVD protection[67,68]. However, androstenediol taken as a dietary supplement was associated with increased LDL and unfavorable CHD risk in men participating in a high-intensity resistance training program[69], pointing to the potential difference between beneficial endogenous effects of a genetic variant that both decreases LDL and increases androstenediol compared to the potential deleterious opposite exogenous effects of androstenediol as a dietary supplement. Second, we found the *SLC22A8/A24* locus that was previously associated with ETIO-G to be associated with higher estriol. Estriol is a weaker form of estrogen, and interestingly, in UKBB, this region was associated with cholecystitis without cholelithiasis (inflamed gallbladder without gallstones). This association may be the underlying inflammatory first step in the process that leads to two-fold increase gallstone formation in women of reproductive age or on birth control medication that have estrogen compared to males[70], and maybe informative in personalized medicine. This association is independent of the nearby *FADS* gene region that has been associated with gallstones[71] and assumed to work through its effect on lipids. However, given the lack of replication in GOLDN, further investigation is warranted.

The phenotype and genotype correlation pattern as well as the heritability estimates in our study were generally in line with other general population studies. This study also replicated many of previously identified common variants which highlight the generalizability of the Amish results to the general population, besides its added value in identifying rare population variants enriched by drift. While traditional lipids explained a small proportion of the variance of the lipidome, and the lipidome explained a significant proportion of the genetic variance in traditional lipids, the overlap was incomplete leaving a significant proportion in both sides remained to be explained. This limited overlap highlights the difference in the genetic architecture and the complimentary value in using both traditional lipids and lipidome in understanding lipid genetic architecture.

While this study may be limited by a relatively small sample size, we were still able to identify three novel rare-population variants. Larger sample size in Amish and other founder population will undoubtedly identify more rare variants which would be challenging to identify in the general population and can inform biological mechanisms and therapeutic targets relevant to all humans. While this GWAS included 355 lipid species, the largest to date, we excluded lipid species with low quality data, so more complete profiling is warranted for comprehensive interrogation.

## Methods

**Study populations.** The Old Order Amish (OOA) population of Lancaster County, PA immigrated to the Colonies from Central Europe in the early 1700's. There are currently around 40,000 OOA individuals in the Lancaster area, nearly all of whom can trace their ancestry back about 15 generations to approximately 750 founders. Investigators at the University of Maryland Baltimore have been studying the genetic determinants of cardiometabolic health in this population since 1993. To date, over 7000 Amish adults have participated in one or more of our studies as part of the Amish Complex Disease Research Program[72]. The samples used in this study were participants of Heredity and Phenotype

Intervention (HAPI) Heart Study[73]. Briefly, HAPI was initiated in 2002 to identify the genetic and environmental determinants of responses (blood pressure, triglyceride excursion and platelet aggregation) to four short-term interventions including a cold pressor stress test, a high salt diet, a high fat challenge, and an aspirin therapy in a four-week time period. HAPI recruited 1003 OOA, and the interventions were carried out in 868 relatively healthy OOA adults (>=20 years of age). Participants were asked to discontinue the use of all medications, vitamins and supplements for at least 7 days prior to the first visit and during the interventions, to fast at least 12 h prior to their visit, and to restrain themselves from doing excessive physical activity on the morning of their appointment. Baseline blood drawn from 650 participants was used for the lipidomic profiling in this study. The study protocol was approved by the institutional review board at the University of Maryland. Informed consent was obtained from each of the study participants.

GOLDN (Genetics of Lipid Lowering Drugs and Diet Network), the largest study of postprandial dyslipidemia that offers NMR, clinical lipid, and lipidomic measures, was initiated to assess the interaction of genetic factors with environmental interventions (intake of a high-fat meal and/or fenofibrate treatment)[74]. Briefly, the study recruited European American families with at least two siblings from two field centers (Minneapolis, MN and Salt Lake City, UT) of the Family Heart Study (FHS). Participants were excluded if they (1) had fasting triglycerides (TGs) ≥ 1500 mg/dL, (2) had a history of kidney, liver, pancreas, or gallbladder disease, recent myocardial infarction or revascularization, or nutrient malabsorption, (3) reported a current use of insulin, and (4) were pregnant or lactating. Of the 1327 participants who were initially screened, 1048 (including 546 women) met the eligibility criteria and were included in the study. A written consent form was provided for each participant and the protocol of the study was reviewed and approved by the institutional review boards at the University of Utah, University of Minnesota, and Tufts University/New England Medical Center.

**Lipidomic profiling.** The technical details of the laboratory protocols and methods of the lipidomics experiments are described in our previous paper[24] and reproduced here for completeness.

Baseline HAPI and GOLDN lipidomics data includes neutral lipids and phospholipids that were collected using UPLC–QTOFMS at the West Coast Metabolomics Center at University of California Davis. The protocol for this measurement was described in detail elsewhere[75,76]. Briefly, the whole process was divided into three steps: lipid extraction and separation, data acquisition and lipid identification. Methyl tert-butyl ether (MTBE), methanol, and water were used to extract plasma lipids. The quality control (QC) samples were method blanks and pooled human plasma (BioreclamationIVT). The separated non-polar phase was injected into a Waters Acquity UPLC CSH C18 (100 mm length × 2.1 mm id; 1.7 μm particle size) with an additional Waters Acquity VanGuard CSH C18 pre-column (5 mm × 2.1 mm id; 1.7 μm particle size) maintained at 65 °C was coupled to an Agilent 1290 Infinity UHPLC (Agilent Technologies) for ESI positive and negative modes. Mobile phase modifiers included ammonium formate and formic acid for positive mode and ammonium acetate (Sigma–Aldrich) for negative mode. The same mobile phase composition of (A) 60:40 v/v acetonitrile:water (LC-MS grade) and (B) 90:10 v/v isopropanol:acetonitrile was used for both positive and negative modes. An Agilent 6550 QTOF with a jet stream electrospray source was employed for acquiring full scan data in the mass range m/z 65–1700 in positive and negative modes with scan rate of 2 spectra/second. Instrument parameters were as follows for the ESI (+) mode – gas temperature 325 °C, gas flow 8 l/min, nebulizer 35 psig, sheath gas temperature 350 °C, sheath gas flow 11, capillary voltage 3500 V, nozzle voltage 1000 V, fragmentor voltage 120 V and skimmer 65 V. In negative ion mode, gas temperature 200 °C, gas flow 14 l/min, fragmentor 175 V, with the other parameters identical to positive ion mode. Data are collected in centroid mode at a rate of 2 scans per second. Injection volume was 1.7 μL for the positive mode and 5 μL for the negative mode. The gradient started at 15% B, ramped to 30% at 2 min, 48% at 2.5 min, 82% at 11 min, 99% at 11.5 min and kept at 99% B until 12 min before ramping down to 15% B at 12.1 min which was kept isocratic until 15 min to equilibrate the column. The total run time was 15 min and the flow rate was 0.6 ml/min. Data were acquired in nine batches and every ten samples, one quality control sample was analyzed. MS1 data were acquired for all samples, and MS/MS data were acquired for a set of pooled samples. Data were processed with the Agilent Quant 7.0 software. Lipids levels were reported as chromatographic peak heights. Stringent quality control was performed for the lipidomics data using eight previously reported measures[77]. We observed very low missing data rate of approximately 0.5% for known profiled lipids in positive ionization mode. The missing data rate was higher for certain molecular weight of the lipid compound and retention behavior. For example for a few low intensity compounds such as high molecular weight triacylglycerols (TG (62:4) or TG (60:6)) the missing data rate was up to 20%. Missing value imputation was performed using our previously developed computational routine[78] before SERRF normalization[78]. After normalization, the relative standard deviation of quality control samples is 4.7% and 3.4% for negative and positive mode respectively. Lipid identification was performed by converting the acquired MS/MS spectra to the mascot generic format (MGF) and then a library search using the in-silico MS/MS library LipidBlast. After quality control, 355 lipid compounds were included in the HAPI lipidomic GWAS and 328 in the GOLDN replication study.

**HAPI chip genotyping and imputation**. Genomic DNA was extracted from whole blood from 1856 individuals of the OOA and quantitated using PicoGreen. Genome-wide genotyping was performed with Affymetrix 500 K ($n = 1252$, including all HAPI participants) and Affymetrix 6.0 ($n = 604$) arrays at the University of Maryland Biopolymer Core Facility. The BRLMM algorithm was used for genotype calling. Prior to imputation, the two chips were merged into a single file. Samples with call rate <0.93, high level of Mendelian error, or gender mismatch were excluded. Variants with >2% missing data, Hardy-Weinberg expectation (HWE) $p$-value < 1E-10, Mendelian errors >1% or with MAF < 0.01 ($N = 366,169$) were excluded. We also excluded variants on the Y chromosome and mitochondrial genome, palindromic variants with frequency >0.4, and variants that were not in the TOPMed Freeze 5b reference panel. These QC procedures left 1,833 participants and 307,238 variants in the genotype file for imputation. The genotype data were uploaded to the Michigan Imputation Server[79] where the pre-phasing was performed using Eagle v2.4[80], and then imputation to the TOPMed Freeze 5b reference panel was performed using Minimac4[48]. Following imputation, we excluded variants with imputation quality/INFO < 0.9, MAF < 0.01 or deviation from HWE at $p < 1.0E-09$. These processes left 7,917,357 variants for the association analysis with 639 samples with both phenotypes and genotypes.

**Whole-genome sequencing for GOLDN**. Whole-genome sequencing (WGS) for the Trans-Omics in Precision Medicine (TOPMed) program was supported by the National Heart, Lung and Blood Institute (NHLBI). WGS for "NHLBI TOPMed: Genetics of Lipid Lowering Drugs and Diet Network" (phs001359) was performed at the North West Genomics Center, University of Washington. Centralized read mapping and genotype calling, along with variant quality metrics and filtering were provided by the TOPMed Informatics Research Center. Data management, sample-identity QC, and general study coordination were provided by the TOPMed Data Coordinating Center. Library preparation and whole-genome sequencing were performed on 967 GOLDN samples by North West Genomics Center, University of Washington. The NHLBI Informatics Resource Core at the University of Michigan performed alignment, base calling, and sequence quality scoring and variant calling of all TOPMed samples using the GotCloud pipeline[81]. Variant calling used a support vector machine (SVM) trained using known variants. Variants passing all quality filters with read depth at least 10 were delivered in BCF format and used for association analysis. Further variant QC included removing all sites in low-complexity regions[82], and on the X chromosome. There were 835 GOLDN samples with both lipidome and WGS data and used for the GWAS.

**Phenotype preparation**. In HAPI, to adjust for potential technical artifacts and non-normality of raw lipidomic values, each lipidomic was first regressed in a linear model adjusting for age, age squared, sex, and experimental technical artifacts including batch, box, row, position and plate, then the regression residuals were inverse normalized. No adjustment for medication was included as none of the HAPI subjects were on lipid lowering medication. These transformed lipidomic values were used in all Amish analyses. The identical procedure was applied to lipid phenotypes, excluding technical artifacts from the linear regression, to standardize analyses combining both lipid and lipidome.

In GOLDN the exact same lipid panel was completed and an inverse rank normal transformation was used on each lipid class phenotypes.

**Variance decomposition**. Mixed model variance component analysis was used to partition observed phenotypic variance $\sigma^2_p$ into causal components $\sigma^2_k$ and residual error $\sigma^2_e$, that is,

$$\sigma^2_p = \sigma^2_1 + \sigma^2_2 + \ldots + \sigma^2_n + \sigma^2_e \qquad (1)$$

The variance components $\sigma^2_k$ correspond to random effects $b_k$ assumed to follow multivariate Gaussian distribution $b_k \sim N(0, \sigma^2_k \Sigma_k)$, with mean zero, covariance matrix $\Sigma_k$. The matrix $\Sigma_k$ contains pairwise covariance values between subjects and the variance components $\sigma^2_k$ are estimated using mixed model maximum likelihood methods incorporating corresponding covariance matrices. For interpretability the estimated variances $\sigma^2_k$ are converted to the proportion of phenotypic variance explained, called $\lambda_k$, by dividing by the phenotypic variance $\sigma^2_p$, that is, $\lambda_k = \sigma^2_k / \sigma^2_p$. Likelihood ratio test (LRT) $p$-values can be used to compare nested models of different random effects to determine if the model with more components provides significantly better fit of the data. The LRT is applied using standard sequential procedures to build the most parsimonious causal component decomposition of the phenotypic variance using a predefined $p$-value threshold of 0.05. At each step LRT $p$-values are computed comparing the current best model with that model plus one of the remaining random effects. The current model is then updated with the remaining random effect with the smallest $p$-value. The procedure is repeated until no LRT $p$-value is less than 0.05.

**Additive and dominant heritability**. The pedigree kinship coefficient measures the expected probability that two subjects share an allele identical by descent given the pedigree structure. An Amish kinship covariance matrix was constructed using a single 14-generation pedigree that connects all 650 subjects back to their 18th century founders. The Amish population structure provides unique opportunities

to separate genetic and environmental effects important in lipidome as many distant relative pairs, such as cousins, share genes from the same founder but not common environments such as diet and lifestyle. A dominance covariance matrix was also constructed using the pedigree structure that measures the probability that two subjects share a genotype identical by descent.

**Data-derived covariance matrices**. In multivariate statistics the sample covariance matrix can be constructed using any set of variables measured across subjects. First consider the design or data matrix X that contains measured variables such as lipidome on subjects that is used in regression to estimate the effect of the variables as fixed effects. To construct the covariance matrix the variables in X are first mean centered and normalized to remove potential scale differences between them. Then the subject-by-subject sample covariance matrix S is defined as S = XX', where X' is the transpose of X. We describe details of how covariance matrices were constructed using genetic markers, lipidomics and lipids.

**Lipidome variance explained by known lipid variants**. To measure the proportion of lipidomic and traditional lipid variance due to genetic markers associated with HDL, LDL, TC and TG lipid levels, genetic relatedness matrices (GRM) were constructed using SNPs identified from the literature[30,37,83–86] as being genome-wide significant for each lipid plus known Amish-specific variants (*APOB*_rs5742904[15], *APOC3*_rs76353203[14], *B4GALT1*_rs551564683[21], *TIMD4*_rs898956003[20]) (4 variants). The number of literature SNPs that were present in the Amish genotype data and used were 99 for HDL, 77 for LDL, 97 for TC, and 67 for TG. This analysis was restricted to 194 lipid species with baseline significant heritability (Kinship + 4 V heritability $p$-value < 0.01). We chose to use (Kinship + 4 V) as the base model in this analysis to focus on the effect of common known lipid variants rather than the strong effect of these 4 Amish-enriched rare-population variants, however, we included *APOB*, *B4GALT1* and *TIMD4* variants in the LDL and TC GRMs and the *APOC3* variant in the TG and HDL GRMs to account for any confounding/residual effect of these variants with common variants, if any. SNP genotyping was available on the 639 subjects with lipidomics. To estimate the genetic contribution of GWAS SNPs associated with lipidomic and traditional lipids as phenotypes a mixed model analysis was performed including kinship and lipid SNP GRM as random effects.

**Genetic and phenotypic correlation**. The software biMM[87] was used to calculate additive genetic correlations between 359 variables (355 lipidomics and 4 traditional lipids (HDL, LDL, TC, TG)) on data from 639 subjects using the Amish kinship matrix. biMM returns bivariate mixed model maximum likelihood estimates of genetic and environmental correlation that includes estimates of heritability of each trait genetic correlation between them allowing for residual errors between traits. biMM does not constrain genetic correlation estimates to be in the range [−1,1], thus out-of-range correlations, which were common when one or both traits have low heritability, were set to missing as estimates were not deemed reliable. There were 7428 with values < −1.0 and 9020 with values > 1.0. Out-of-range estimates are represented by white squares in the heatmap, and only 64,621 correlation are included in Supplementary Data 3. R[88] was used to calculate the pairwise phenotypic Pearson correlations for lipid species and traditional lipids.

**Traditional lipid variance explained by lipidome classes**. Covariance matrices were constructed for each of the 13 lipid classes (ACT, CE, Cer, DAG, FA, GlcCer, LPC, LPE, PC, PE, PI, SM, TAG) using mean centered and normalized raw lipidomic values from each class in the data matrix X. These covariance matrices were used in a mixed model with traditional lipid (HDL, LDL, TG and TC) as the trait and kinship and lipidomic class as random effects. We estimated the marginal contribution of each lipid class separately and also performed a sequential analysis as described above to determine the best multi-class model fit for each lipid which estimates the joint proportion of traditional lipid variance accounted for by the lipidomic class.

**Association analyses**. In HAPI, genetic association analysis of inverse normalized lipid species was performed using linear mixed models to account for familial correlation using the genetic relationship matrix (GRM) as implemented in the Mixed Model Analysis for Pedigree and population (MMAP)[89]. For 180 lipid species that showed nominal association ($p < 1.0E-03$) with any of the 4 Amish enriched lipid variants (APOB_rs5742904, APOC3_rs76353203, B4GALT1_rs551564683, TIMD4_rs898956003), we reran the association analyses adjusting for the variant(s) as detailed in Supplementary Data 2. The effect size for all traits is reported in standard deviation units for comparability. Multiple testing adjusted significance threshold of 4.5E-10 was determined by dividing the standard GWAS level of 5E-08 by the number of principle components (110) that explained >95% of the variance in the 355 metabolomic variables. All associations between 5E-08 and 4.5E-10 were considered suggestive. The number of independent signals at each locus was determined using sequential conditional analysis. The novel loci were determined by conditioning on preidentified variants within 1 Mb from the top associated variant.

In GOLDN we performed a parallel linear mixed model analysis on the inverse normally transformed lipid phenotypes in Saige-0.29 pipeline deployed in Encore

analytics framework (i.e. Fast linear mixed model with kinship adjustment (saige-qt)). Pre-derived top 10 PCs from TOPMed WGS cohort was adjusted as covariates along with age, sex and center.

Bonferroni corrected p-value of 1.3E-05 was used for GOLDN replication accounting for 3631 trait-variant pairs of GOLDN association results included in Supplementary Data 6

**Annotation and biobank lookups**. Look ups of top results in publicly available PheWAS databases including UK Biobank[90–93], FinnGen[94] and BioBank Japan[95] was performed using the "Omics Analysis, Search and Information System" (OASIS)[96], a web-based application for mining and visualizing GWAS results via integration with a broad spectrum of available databases for functional annotation such as dbSNP[97], gnomAD[98], GTEx[99], Open Targets Genetics[100], eigenPC[40], RegulomeDB[41], The ensembl regulatory build[101] chromatin state in four different tissue[42], and the UCSC Genome Browser[102] to visualize their proximity to functional regions (e.g. binding sites, Dnase hypersensitivity sites, enhancer/promoter regions).

**Reporting summary**. Further information on research design is available in the Nature Research Reporting Summary linked to this article.

## Data availability

Amish phenotype, genotype, and lipidome data are available to academic investigator via a Data Use Agreement with UMB. GOLDN phenotype, genotype, and lipidome data are available to academic investigator via a Data Use Agreement with UAB. Please contact the corresponding author to initiate the data request. Source data underlying Figs. 1b and 2 are presented in Supplementary Data 2. Summary statistics are available through the GWAS Catalog (https://www.ebi.ac.uk/gwas/home) under accession GCP000299.

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

## Acknowledgements

We gratefully thank the Amish community, the Amish Research Clinic staff and liaisons, and the participants of the GOLDN study. We also thank Simeon I. Taylor for insightful suggestions, comments and discussions. The HAPI Heart study was supported by U01HL072515. The analysis methods and software were supported by U01HL084756 and U01HL137181. The GOLDN study was supported by the NHLBI grant U01HL072524-04 and R01HL091357.

## Author contributions

Conceived, designed and supervised the work: M.E.M., J.R.O., M.R.I., D.K.A., H.K.T. Results interpretation: M.E.M., J.R.O., A.L.B., M.R.I., D.K.A., S.A. Manuscript preparation: M.E.M., J.R.O. Provided samples, genotype and phenotype data: A.R.S. Data analyses; M.E.M., J.R.O., S.A., V.S., J.P., K.A.R., M.B., A.P. Performed lipidomic profiling and processed the raw data; T.K., D.K.B., S.F. All authors read, revised and approved the paper.
