## [Peer Review File · Communications Biology]

Reviewers' comments:

Reviewer #1 (Remarks to the Author):

The manuscript reports on a genome-wide association study of 355 circulating lipids in 650 Amish individuals. The authors identify 3 Amish-enriched loci and attempt to replicate and finemap the causal variants at these loci using the outbred GOLDN study. The work demonstrates the power of founder populations to identify novel genes (and potential drug targets) for lipids and lipid disorders.

Overall, this is a solid and novel contribution. The analyses are expertly conducted and the data appear convincing despite the lack of independent replication (challenging in this context). It is important to note that the GOLDN sample provides exclusion evidence rather than replication.

In two of the 3 identified loci, the authors identify variants changing the amino-acid sequence as their best candidate for causal variant. The third locus prioritization scheme is unclear. It is unclear why the authors reduce their candidate variants (from 202 to 27) solely based on p-value and LD with the top variant. While they cite functional annotation as a reason to retain the intronic variant as the putatively causal, there is no information how functional annotation was used in the overall strategy. Please provide additional information on what and how functional annotation was used.

While the data support well the conclusions, additional supporting evidence could be gathered that would further enhance the value of the manuscript. Notably, look-up of the 3 variants in the WGS analysis of traditional lipids conducted in TOPmed in over 60,000 subjects may be of further interest.

Additionally connecting these novel variants or genes to clinical endpoints would also be of interest.

Minor point: Please additional information regarding the analyses examining the genetic contribution of traditional lipids to lipidome. Which variants were used?

Reviewer #2 (Remarks to the Author):

The study of Motasser and al is very elegant and well designed. The manuscript is very clear and well-written. This study aimed at identifying new rare genetic determinants of the lipidome by taking advantages of the genetic structure of an Amish founder population. This is a very interesting approach leveraging unique access to such population. The authors performed a GWAS of 355 lipid species in 650 individuals from the Old Order Amish founder, and then tentatively replicated their findings in an independent population-based cohort. As main findings, they identified 2 new variants in APOB and APOC3 associated with lipid species, and 3 variants associated with several sphingolipids, and they proposed potential functional/causal variants in each locus identified. They also identified 7 previously known common loci that reinforced the robustness of their method, along novel associations with two sterols: androstenediol with UGT locus and estriol with SLC22A8/A24 locus.

Overall, the authors applied well-known methods appropriate for genetic studies in families from founder population under the hypothesis that such population has the power to detect new variants that are rare or absent in the general population. I do not have any specific concerns regarding the methods, which have been clearly explained, and previously validated. The results and tables/figures are also well presented, as we expect for such genetic studies.

My few suggestions are as follows:

- This could have been interesting to estimate the genetic correlations of the metabolites with cardiovascular traits, in addition to the genetic correlations conducted between metabolites and lipid particles (LDL-C, ...)
- The potential pleiotropic effects of identified loci should have been investigated and discussed, particularly if future Mendelian Randomization analyses will use these genetic results as inputs.

Reviewer #3 (Remarks to the Author):

In this study, Montasser et al. have performed a genome-wide association analysis to identify genetic variants associated with plasma lipidome comprising of 355 lipid species in Amish population. Authors have also attempted to replicate the findings in an independent cohort with similar lipidomic profiling. Authors further looked up for the association of identified variants with disease outcomes in publicly available resources. The main strengths of the manuscript are extensive lipidomic profiling with many new lipid species that are not reported previously and founder population which provide unique opportunity to identify new genetic variants/loci. Altogether, this is an interesting study given the limited number of studies in the field. My concerns and suggestions for the manuscript are listed below:

1. The sample size is relatively small compared to the similar studies in lipidomics and metabolomics. Though the founder population provide higher statistical power to capture association of highly enriched variants, the study seems to be under-powered to detect associations with moderate or small effect sizes.
2. As independent dataset (GOLDN) with similar lipidome profiling was available, I was expecting that authors would have performed meta-analysis to increase the statistical power. Meta-analysis of all variants with suggestive association in Amish population with that of GOLDN could provide more genetic variants with significant associations.
3. Presentation of results for heritability estimates and variance explained are bit confusing. Are the heritability estimates for "kinship" calculated in the same way in Figure 1 and 2. If so, the estimates seem to differ in the two figures. It is difficult to interpret the variance explained by known variants from the Figure 2 and should be provided in supplementary table for clarity.
4. Result section on "Lipidome contribution to traditional lipids": It is not clear what information does this analysis provide. Maybe it would be helpful if authors could discuss the motivation for this analysis and their interpretation in more detail.
5. Authors have mentioned that "Leveraging results from the GOLDN study we were able to finemap large numbers of variants present on long Amish-enriched haplotypes to identify a potentially functional variant in a biologically plausible gene for each of the three loci." This claim is not supported by the study results as none of the variants from these three loci was associated or available in GOLDN. Moreover, authors conclusion that if the variants are not replicated in another cohort, they are not causal, is not correct. As stated in "The remaining 391 variants (MAF: 0.001- 0.48) were non-significant in GOLDN, providing confidence they are non-causal." There could be many various reasons for non-replication of variants in another cohort such as statistical power, differences in genetic architecture, allele frequencies, false positives.
6. Results of the biobank lookups for the identified variants are not discussed in the manuscript. How many of the novel identified variants or loci were associated with disease outcomes in biobank cohorts?
7. What quality control measures were applied to the lipidome data? What was the missing rate?
8. "The position of this rare missense variant rs536055318 (A263T) in an aTSS within the promoter region of GLTPD2 can alter its expression". Is this variant an eQTL for GLTPD2? The information could be obtained from publicly available databases such as GTEx.
9. Introduction section is very limited. It would be useful for better understanding of the importance and relevance of the study to provide more detailed background information about lipidome, and Amish founder population.
10. It is not mentioned in the method section, what tool or statistical package was used to perform the GWAS.

We would like to thank the reviewers for their thorough evaluation and insightful comments that have helped us to further improve our manuscript. Here, we provide a point-by-point response to the referees' comments

Reviewers' comments:

Reviewer #1 (Remarks to the Author):

The manuscript reports on a genome-wide association study of 355 circulating lipids in 650 Amish individuals. The authors identify 3 Amish-enriched loci and attempt to replicate and finemap the causal variants at these loci using the outbred GOLDN study. The work demonstrates the power of founder populations to identify novel genes (and potential drug targets) for lipids and lipid disorders.

Overall, this is a solid and novel contribution. The analyses are expertly conducted and the data appear convincing despite the lack of independent replication (challenging in this context). It is important to note that the GOLDN sample provides exclusion evidence rather than replication.

In two of the 3 identified loci, the authors identify variants changing the amino-acid sequence as their best candidate for causal variant. The third locus prioritization scheme is unclear. It is unclear why the authors reduce their candidate variants (from 202 to 27) solely based on p-value and LD with the top variant.

There are several approaches to fine mapping. Conditional analysis is useful to determine the number of independent loci by adding sentinel SNPs to the regression until p-values of the non-sentinel SNPs fall below a pre-determined significance threshold. The number of sentinel SNPs required is the number of independent loci. Our conditional analysis showed that there is only one signal in this locus. PAINTOR is formal fine mapping approach that incorporates functional information in addition to LD. PAINTOR results using 202 or 27 variants, different input parameters and functional information consistently identified the top associated variant (rs7863920, $p=6.2E-18$) as having the highest posterior probability of causality.

We have extensive experience with disentangling LD-driven signals on long-range haplotypes in the Amish to identify the best causal candidate for medium-to-large effect rare alleles. The best candidate is generally the top signal and historical recombination around the causal variant will decrease the $-\log_{10}$ p-value of flanking SNPs quickly given the effect size. Generally SNPs with one or two decimal place difference can be excluded. This statistical approach correctly identified several rare population causal alleles in the Amish that were later functionally confirmed: *APOB*_rs5742904¹, *APOC3*_rs76353203², *B4GALT1*_rs551564683³, *TIMD4*_rs898956003⁴.

Although we applied a relatively more relaxed criteria in this study to select the top candidates (p -values $< 8.5E-16$ and $r^2 > 0.75$) at the third locus, the proposed candidate has the statistical characteristics of being the causal variant. However, we cannot rule out the possibility that some other variant that may failed genotyping or imputation is the real causal variant driving this

association signal, in which case, we would expect that variant to have similar or better p-value than our current candidate. Finally, inferring causality requires functional validation.

While they cite functional annotation as a reason to retain the intronic variant as the putatively causal, there is no information how functional annotation was used in the overall strategy. Please provide additional information on what and how functional annotation was used.

We look up variant functional annotations using the “Omics Analysis, Search and Information System” (OASIS)⁵, a web-based application for mining and visualizing GWAS results via integration with a broad spectrum of available data bases such as dbSNP⁶, gnomAD⁷, GTEx⁸, Open Targets Genetics⁹, eigenPC¹⁰, RegulomeDB¹¹, The ensembl regulatory build¹² chromatin state in four different tissues¹³, and the UCSC Genome Browser¹⁴ to visualize their proximity to functional regions (e.g. binding sites, Dnase hypersensitivity sites, enhancer/promoter regions).

For the chromosome 9 locus we looked at all available functional annotation for the top 27 variants (figure 1), and found that this intronic variant stands out with the best functional annotation including the top ENCODE DNase score of 1000 indicating very strong evidence of a DNase I hypersensitivity site¹⁵, an eigenPC score of 3.5 (top 1%) indicating a strong functional prediction based on conservation and allele frequency¹⁰, and is predicted to affect transcriptional factor binding with a 2a RegulomeDB classification¹¹. The variant is located in a weak transcription site in the islet and skeletal muscle, in a genic enhancer region in liver tissue, and in an active enhancer region in adipose tissue¹³. The functional annotation provides significant support to the strong statistical evidence that the variant is our current best causal candidate, with the same caveats that the true variant may not be genotyped and that any candidate will require functional validation to prove causality. We clarified this in the revised manuscript.

Figure 1: Functional annotation for the top 27 variants in chromosome 9 locus highlight the intronic variant (rs531892793, p=3.9E-17) as a strong potentially functional variant in this region. r^2 , D' : r^2 and D' prime correlation with the top variant. RglmDB: RegulomeDB classification¹¹, eigenPC score¹⁰, Dnase: DNase I hypersensitivity site¹⁵, Reg: The ensembl regulatory build¹², and chromatin state in four different tissues¹³ (islet, liver, adipose, and skeletal muscle¹³). 5.Tx (Strong_transcription), 6.TxWk (Weak_transcription), 8.EnhG (Genic_Enhancer1), 9.EnhA1 (Active_Enhancer1), 10.EnhA2 (Active_Enhancer1), 11.EnhWk (Weak_Enhancer), 16.RPC (Repressed_PolyComb), 17.RPCWk (Weak_Repressed_PolyComb), 18.Quies (Quiescent/low).

	pVal_SNP	r^2	D'	RglmDB	eigenPC	Dnase	Reg	islets	liver	adipose	sklms
rs7863920	6.22E-18			7	-0.257			18.Quies	17.RPCWk	18.Quies	18.Quies
rs138715201	9.37E-18	0.79	0.94	6	-0.251			18.Quies	18.Quies	6.TxWk	18.Quies
rs140093477	9.37E-18	0.79	0.94	6	-0.158			6.TxWk	6.TxWk	6.TxWk	6.TxWk
rs146764160	9.37E-18	0.79	0.94	4	0.105	371	open	16.RPC	18.Quies	16.RPC	16.RPC
rs186979747	9.37E-18	0.79	0.94	6	-0.139			5.Tx	5.Tx	8.EnhG	5.Tx
rs542505754	9.37E-18	0.79	0.94	7	-0.136			6.TxWk	6.TxWk	11.EnhWk	6.TxWk
rs952789069	1.10E-17	0.79	0.94	7	-0.094			17.RPCWk	18.Quies	17.RPCWk	17.RPCWk
rs144332784	1.24E-17	0.86	0.98	7	-0.218			17.RPCWk	17.RPCWk	17.RPCWk	18.Quies
rs190844068	1.24E-17	0.86	0.98	6	-0.168			17.RPCWk	17.RPCWk	18.Quies	17.RPCWk
rs185511591	1.65E-17	0.86	0.98	7	-0.198			18.Quies	18.Quies	18.Quies	18.Quies
rs566604550	1.65E-17	0.86	0.98	6	-0.284			18.Quies	18.Quies	18.Quies	18.Quies
rs181234756	1.69E-17	0.86	0.98	7	-0.220			17.RPCWk	17.RPCWk	18.Quies	18.Quies
rs974137970	1.69E-17	0.86	0.99	5	-0.177			17.RPCWk	6.TxWk	18.Quies	18.Quies
rs927992688	1.69E-17	0.86	0.98	6	-0.152			18.Quies	18.Quies	18.Quies	18.Quies
rs146981196	1.69E-17	0.86	0.98	6	-0.012			18.Quies	17.RPCWk	18.Quies	18.Quies
rs141683313	1.71E-17	0.83	0.95	6	-0.108			18.Quies	6.TxWk	6.TxWk	6.TxWk
rs1014518150	1.76E-17	0.83	0.95	6	-0.173			17.RPCWk	17.RPCWk	6.TxWk	18.Quies
rs533456288	3.83E-17	0.78	0.93	5	-0.175	214		6.TxWk	18.Quies	18.Quies	17.RPCWk
rs543685197	3.86E-17	0.78	0.93	5	-0.039	313		6.TxWk	18.Quies	18.Quies	17.RPCWk
rs531892793	4.00E-17	0.78	0.93	2a	3.501	1000	proFR	6.TxWk	8.EnhG	9.EnhA1	6.TxWk
rs956743578	4.12E-17	0.78	0.93	5	-0.009			6.TxWk	5.Tx	6.TxWk	6.TxWk
rs528148931	4.15E-17	0.76	0.93	5	-0.140			10.EnhA2	9.EnhA1	6.TxWk	11.EnhWk
rs993853922	4.15E-17	0.78	0.93	5	0.018			6.TxWk	5.Tx	6.TxWk	6.TxWk
rs554319484	4.23E-17	0.86	0.98	7	-0.211			17.RPCWk	16.RPC	18.Quies	18.Quies
rs978182891	4.23E-17	0.86	0.98	7	-0.209			17.RPCWk	16.RPC	18.Quies	18.Quies
rs1011013571	1.36E-16	0.82	0.95	7	-0.145			17.RPCWk	17.RPCWk	6.TxWk	6.TxWk
rs78967418	8.59E-16	0.76	0.87	6	-0.205			17.RPCWk	17.RPCWk	6.TxWk	18.Quies

While the data support well the conclusions, additional supporting evidence could be gathered that would further enhance the value of the manuscript. Notably, look-up of the 3 variants in the WGS analysis of traditional lipids conducted in TOPmed in over 60,000 subjects may be of further interest.

As these 3 variants are rare in general outbred populations, we opted to use the UKBB, which has at least a 7-fold greater sample size of European ancestry than TOPMed (N = ~500K vs ~60K) and whose summary results are publicly available for look up of traditional lipids (Supplementary Table 6, columns V). *GLTPD2_rs536055318* (MAF=0.001 in general European populations), was associated with lower levels of triglycerides (TG) with a suggestive p-value of 6.9E-08. The association with lower LDL-C, lower total cholesterol and higher HDL-C were much less significant (p-value = 0.03, 0.06 and 0.02, respectively). Of note that it required 461,140 UKBB subjects to find this suggestive association with TG, so TOPMed sample size of 60K would not have any power to detect this association.

The other 2 variants (*CERS5_rs771033566*, and *AKNA_rs531892793*) are extremely rare (MAF= 0.0001 and 0.000001 in general European population) and were not reported in UKBB.

There was no associations for these 2 variants with lipids (p-value = 0.2-0.9) in the 1,083 Amish samples included in TOPMed. As there are only a few sporadic copies in the non-Amish TOPMed the associations would not change in total TOPMed

Additionally connecting these novel variants or genes to clinical endpoints would also be of interest.

We agree with the reviewer that connecting the variants with clinical endpoints would be interesting. Unfortunately clinical endpoints are not available in this Amish population, but we looked up all top results in publicly available PheWAS databases including UK Biobank¹⁶⁻¹⁹, FinnGen²⁰ and BioBank Japan²¹ (Supplementary Table 6, columns T-X), and curated previous publication.

The *GLTPD2*_rs536055318 variant was associated with numerous seemingly independent phenotypes such as tooth disorder (eruption and development), allergic rhinitis, calculus of the kidney, allergic reaction to food, type of cooking fat/oil, very low fat olive spread, granulomatous disorders of skin and prurigo nodularis, but we found no clear link to CVD in this list. However, a Finnish-enriched *GLTPD2* intronic variant (rs79202680) has been recently associated with reduced atherosclerosis²², pointing to a potential gene-CVD link.

The other two variants *CERS5*_rs771033566, and *AKNA*_rs531892793) are extremely rare (MAF= 0.0001 and 0.000001 in general European population) and are not reported in UKBB nor any other publicly available pheWAS database.

Minor point: Please additional information regarding the analyses examining the genetic contribution of traditional lipids to lipidome. Which variants were used?

The purpose of examining the genetic contribution of traditional lipids to lipidome is to understand the relative contribution of each traditional lipid to lipid species and the interplay between components to gain insight into their architecture. Our analysis showed that traditional lipids explained a small proportion of the variance of the lipidome, but that the lipidome explained a significant proportion of the genetic variance in traditional lipids. This analysis confirmed our general expectation that traditional lipids are comprised of lipid species, but that lipid species have a complex architecture independent of traditional lipids. This limited overlap highlights the difference in the genetic architecture and the complimentary value in using both traditional lipids and lipidome in understanding lipid genetic architecture.

Examples of similar variance components analyses in the genetic arena include decomposing phenotypic variance into contributions due to rare and common variants or contributions defined by functional annotation such as promoter, enhancer, etc.

To measure the proportion of lipidomic and traditional lipid variance due to genetic markers associated with HDL, LDL, TC and TG lipid levels, genetic relatedness matrices (GRM) were constructed using SNPs identified from the literature²³⁻²⁸ as being genome-wide significant for each lipid plus known Amish-specific variants (*APOB*_rs5742904¹, *APOC3*_rs76353203², *B4GALT1*_rs551564683²⁹, *TIMD4*_rs898956003⁴) (4 variants). The number of literature SNPs that were present in the Amish genotype data and used were 99 for HDL, 77 for LDL, 97 for TC,

and 67 for TG. *APOB*, *B4GALT1* and *TIMD4* variants were included in the LDL and TC GRMs and the *APOC3* variant in the TG and HDL GRMs.

Reviewer #2 (Remarks to the Author):

The study of Motasser and al is very elegant and well designed. The manuscript is very clear and well-written. This study aimed at identifying new rare genetic determinants of the lipidome by taking advantages of the genetic structure of an Amish founder population. This is a very interesting approach leveraging unique access to such population. The authors performed a GWAS of 355 lipid species in 650 individuals from the Old Order Amish founder, and then tentatively replicated their findings in an independent population-based cohort. As main findings, they identified 2 new variants in *APOB* and *APOC3* associated with lipid species, and 3 variants associated with several sphingolipids, and they proposed potential functional/causal variants in each locus identified. They also identified 7 previously known common loci that reinforced the robustness of their method, along novel associations with two sterols: androstenediol with *UGT* locus and estriol with *SLC22A8/A24* locus. Overall, the authors applied well-known methods appropriate for genetic studies in families from founder population under the hypothesis that such population has the power to detect new variants that are rare or absent in the general population. I do not have any specific concerns regarding the methods, which have been clearly explained, and previously validated. The results and tables/figures are also well presented, as we expect for such genetic studies.

Thank you

My few suggestions are as follows:

- This could have been interesting to estimate the genetic correlations of the metabolites with cardiovascular traits, in addition to the genetic correlations conducted between metabolites and lipid particles (LDL-C, ...)

We agree with the reviewer but we feel that to exhaustively investigate the role of lipidomics with CVD related phenotypes in our database warrants a separate manuscript and beyond the scope this paper whose primary focus is to discover genetic determinants of lipid species. We included traditional lipids, which indeed are CVD phenotypes, because of their direct relevance to lipid species under investigation.

The potential pleiotropic effects of identified loci should have been investigated and discussed, particularly if future Mendelian Randomization analyses will use these genetic results as inputs.

It's our understanding that pleiotropy refers to association with multiple independent phenotypic domains^{30, 31}. Here we have two SNPs that are significantly associated with only one species

and the rest are associated with several highly correlated species, so the multiple association is just a reflection of the strong correlation between several traits within the same domain rather than true pleiotropy between different traits/conditions that reflect different underlying biology.

GLTPD2_rs536055318 was strongly associated with lower level of SM(d40:0) ($p = 1.1E-12$) and suggestively associated with SM(d36:0, d38:0), which reflects the strong phenotypic correlation ($r = 0.72 - 0.87$) between them.

CERS5_rs771033566 was significantly and suggestively associated with lower levels of SM(d32:2) and SM(d30:1), respectively

AKNA_rs531892793 was strongly associated with lower levels of all tested glucosylceramide species (GlcCer(d38:1), (d40:1), (d41:1), (d42:1), (d42:2)) except the one with the shortest acyl chain (GlcCer(d34:1)), which reflects the strong phenotypic correlation between the first 5 ($r = 0.6 - 1.0$) compared to their much lower correlation with GlcCer(d34:1) ($r < 0.2$).

The other 2 known variants, the missense variant R19X (rs76353203) in the *APOC3* gene was significantly associated with lower levels of 3 phosphatidylethanolamines (PE(36:2), (38:6), (34:2)) and suggestively associated with lower levels of another PE, one di- and three triglyceride species. The well-established Amish-enriched familial hypercholesterolemia (FH) variant R3527Q (rs5742904) in the *APOB* gene was significantly associated with increased levels of several cholesterol esters, sphingolipids and phospholipids while there was no association with acylcarnitine, fatty acids, sterols, and glycerolipids. These associations were consistent with the well-known association of these 2 variants with TG/HDL and LDL/TC respectively and as expected based on the HDL and LDL structure and function, and also reflect the correlation between these species (Supplementary table 3)

Reviewer #3 (Remarks to the Author):

In this study, Montasser et al. have performed a genome-wide association analysis to identify genetic variants associated with plasma lipidome comprising of 355 lipid species in Amish population. Authors have also attempted to replicate the findings in an independent cohort with similar lipidomic profiling. Authors further looked up for the association of identified variants with disease outcomes in publicly available resources. The main strengths of the manuscript are extensive lipidomic profiling with many new lipid species that are not reported previously and founder population which provide unique opportunity to identify new genetic variants/loci. Altogether, this is an interesting study given the limited number of studies in the field. My concerns and suggestions for the manuscript are listed below:

1. The sample size is relatively small compared to the similar studies in lipidomics and metabolomics. Though the founder population provide higher statistical power to capture association of highly enriched variants, the study seems to be under-powered to detect associations with moderate or small effect sizes.

We agree that the sample size is smaller than some other publications but not all of them. As the reviewer notes our power is in rare medium-to-large effect alleles enriched in the Amish, and indeed we were able to identify 5 rare population Amish-enriched variants with effect sizes 1.2, -

1.1, -1.5, -1.1, and -0.7. But we also had sufficient power to detect the 7 known common variants with smaller effect sizes (0.4, -0.4, -0.7, 0.98, 0.48, -0.69, and 0.86). So, while our study may be underpowered for identifying common variants with small effects, it is very powerful in identifying rare population variants with much larger effect, and nicely complements other general population studies.

2. As independent dataset (GOLDN) with similar lipidome profiling was available, I was expecting that authors would have performed meta-analysis to increase the statistical power. Meta-analysis of all variants with suggestive association in Amish population with that of GOLDN could provide more genetic variants with significant associations.

At the initial phase of discussing project design we weighed the choice between meta-analysis vs. discovery in Amish and replication in GOLDN. We opted for the discovery-replication design because we expected Amish discovery would be driven by rarer alleles so that even if our primary variant were not present in GOLDN, (non) replication results of Amish variants segregating on the rare variant haplotype could be informative for fine mapping. Meta-analysis would mostly identify already known common alleles but as the review noted the power of our combined sample would not be significantly more informative than existing studies. Both designs have merit, but replication restricted to top hits has reduced the multiple testing burden and we have found that replication is often seen as an essential component to publication.

3. Presentation of results for heritability estimates and variance explained are bit confusing. Are the heritability estimates for "kinship" calculated in the same way in Figure 1 and 2. If so, the estimates seem to differ in the two figures. It is difficult to interpret the variance explained by known variants from the Figure 2 and should be provided in supplementary table for clarity.

The heritability estimates in Figure 1 was performed for all 355 lipid species tested in this study with and without the 4 Amish-enriched variants (*APOB*_rs5742904¹, *APOC3*_rs76353203², *B4GALT1*_rs551564683³, *TIMD4*_rs898956003⁴). Figure 2 shows joint estimates with the lipid GRM added to the base model (Kinship + 4V) for only 194 lipid species that showed p value < 0.01 in Figure 1 for (Kinship + 4V) heritability, since if the heritability is close to zero the variance estimate of the lipid GRM is not robust. This data is now included in Supplementary Table 2, columns J-M, and the labels for Figure 2 was clarified to avoid confusion.

We chose to use (Kinship + 4V) as the base model in this analysis to focus on the effect of common known lipid variants rather than the strong effect of these 4 Amish-enriched rare-population variants, however, we included *APOB*, *B4GALT1* and *TIMD4* variants in the LDL and TC GRMs and the *APOC3* variant in the TG and HDL GRMs to account for any confounding/residual effect of these variants with other common variants, if any.

4. Result section on "Lipidome contribution to traditional lipids": It is not clear what information does this analysis provide. Maybe it would be helpful if authors could discuss the motivation for this analysis and their interpretation in more detail.

The motivation of the variance components analysis is to understand the relative contribution of each lipid species to traditional lipid and the interplay between components to gain insight into their architecture. Our analysis showed that traditional lipids explained a small proportion of the variance of the lipidome, but that the lipidome explained a significant proportion of the genetic variance in traditional lipids. This analysis confirmed our general expectation that traditional lipids are comprised of lipid species, but that lipid species have a complex architecture independent of traditional lipids. This limited overlap highlights the difference in the genetic architecture and the complimentary value in using both traditional lipids and lipidome in understanding lipid genetic architecture. We updated the revised manuscript to clarify the motivation for this analysis.

Examples of similar variance components analyses in the genetic arena include decomposing phenotypic variance into contributions due to rare and common variants or contributions defined by functional annotation such as promoter, enhancer, etc.

5. Authors have mentioned that “Leveraging results from the GOLDN study we were able to finemap large numbers of variants present on long Amish-enriched haplotypes to identify a potentially functional variant in a biologically plausible gene for each of the three loci.” This claim is not supported by the study results as none of the variants from these three loci was associated or available in GOLDN.

The reviewer is correct that we cannot make statistical inferences about variants not in GOLDN, but we can make inferences about being causal candidates.

GOLDN study was used for fine mapping the long Amish haplotype via taking advantage of the difference in the LD structure between the general population and the founder population (Trans-ethnic fine mapping). The difference in LD structure breaks the long haplotype and reduce the number of the credible set by excluding the non-causal variants, leaving a smaller number of variants which help to zero in the causal among them by functional annotation. The variant we proposed as the most likely causal in this region is our best guess given all available data and pending confirmation with functional experiment. However we cannot rule out the possibility that some other variant that may have failed genotyping or imputation is the real causal variant driving this association signal.

The same population genetics model has repeated itself in the Amish for at least a dozen rare variants that were known or shown to be causal over the last 15 years. In 2008 the APOC3 R19X was discovered in the Amish². Statistical evidence overwhelmingly supported causality but as it had not been observed in human outside the Amish, no replication was possible. It took several years to discover other copies. Now we know it is a rare population allele found in most European populations³²⁻³⁵. The discovery led to the development of the antisense drug Volanesorsen by Ionis Pharmaceuticals for treatment of Familial Partial Lipodystrophy³⁶.

Moreover, authors conclusion that if the variants are not replicated in another cohort, they are not causal, is not correct. As stated in “The remaining 391 variants (MAF: 0.001- 0.48) were non-significant in GOLDN, providing confidence they are non-causal.” There could be

many various reasons for non-replication of variants in another cohort such as statistical power, differences in genetic architecture, allele frequencies, false positives.

It is true that there could be many various reasons for non-replication of variants in another cohort such as statistical power, differences in genetic architecture, allele frequencies, false positives, however in the case of APOB, the missense variant rs5742904 is well-established as a causal variant in this region for familial hypercholesterolemia (FH)^{37; 38}, which is highly enriched in the Amish¹. We use this as an example to illustrate a phenomenon that we observe very often in the Amish founder population where a large number of variants on the same long haplotype show significant association due to their strong LD with only one causal variant, that can be identified by testing the association in general population that lack this strong LD and hence can exclude many unlinked variants (Trans-ethnic fine mapping). We use this strategy to identify the most likely causal variant in each locus given all available data and pending confirmation with functional experiment. However, we cannot rule out the possibility that some other variant that may have failed genotyping or imputation is the real causal variant driving the association signal.

6. Results of the biobank lookups for the identified variants are not discussed in the manuscript. How many of the novel identified variants or loci were associated with disease outcomes in biobank cohorts?

We looked up all top results in publicly available PheWAS databases including UK Biobank¹⁶⁻¹⁹, FinnGen²⁰ and BioBank Japan²¹ (Supplementary Table 6, columns T-X), and curated previous publication.

The *GLTPD2*_rs536055318 variant was associated with numerous seemingly independent phenotypes such as tooth disorder (eruption and development), allergic rhinitis, calculus of the kidney, allergic reaction to food, type of cooking fat/oil, very low fat olive spread, granulomatous disorders of skin and prurigo nodularis, but we found no clear link to CVD in this list. However, a Finnish-enriched *GLTPD2* intronic variant (rs79202680) has been recently associated with reduced atherosclerosis²², pointing to a potential gene-CVD link.

The other 2 variants (*CERS5*_rs771033566, and *AKNA*_rs531892793) are extremely rare (MAF= 0.0001 and 0.000001 in general European population) and were not reported in UKBB.

7. What quality control measures were applied to the lipidome data? What was the missing rate?

Stringent quality control was performed for the lipidomics data using eight previously reported measures³⁹

“Quality control was assured by (i) randomization of the sequence, (ii) injection of 10 pool samples to equilibrate the LC–MS system before actual sequence of samples; (iii) injection of pool samples at the beginning and the end of the sequence and between each 10 actual samples, (iv) injection of NIST SRM 1950 at the beginning of the sequence and after injection of 100 actual samples; (v) procedure blank analysis, (vi) replicate analysis of 10 blood plasma samples (covering both the extraction and LC–MS analysis), (vii) checking the

peak shape and the intensity of spiked internal standards and the internal standard added prior to injection, and (viii) monitoring mass accuracy of internal standards during the run”³⁹

The missing data rate depends on multiple factors, the instrument type, the data processing method and further statistical treatment such as gap-filling and imputation of values.

For the pure raw data, we observed very low missing data rate of approximately 0.5% for known profiled lipids in positive ionization mode. The missing data rate was higher for certain molecular weight of the lipid compound and retention behavior. For example for a few low intensity compounds such as high molecular weight triacylglycerols (TG (62:4) or TG (60:6)) the missing data rate was up to 20%.

Missing values are a common occurrence in lipidomics and are usually treated by imputation of a noise value or a minimum value or standard deviation to allow further statistical treatment. Missing value imputation was performed using our previously developed computational routine⁴⁰ before SERRF normalization. We updated the revised manuscript to clarify the QC procedures.

8. “The position of this rare missense variant rs536055318 (A263T) in an aTSS within the promoter region of GLTPD2 can alter its expression”. Is this variant an eQTL for GLTPD2? The information could be obtained from publicly available databases such as GTEx.

The rare missense variant rs536055318 (A263T) has to date not been reported as an eQTL for any gene in any tissue in GTEx⁴¹ and eQTLGen⁴² due to both the rarity of this variant in the general population (MAF 0.001 vs Amish MAF=0.07) and the relatively small sample sizes that exist with expression data on tissues. For example, GTEx has only 208 liver samples with genotypes, which has an expected count of zero copies. Several thousand samples from the general population would be needed to find sufficient copies for regression. The variant is simply too rare for eQTL discovery with current sample sizes.

9. Introduction section is very limited. It would be useful for better understanding of the importance and relevance of the study to provide more detailed background information about lipidome, and Amish founder population.

We limited the length of the introduction section to be able to keep the full length of the text within the word limit. In the revised manuscript we included more information about lipidome, and Amish founder population in the introduction section.

10. It is not mentioned in the method section, what tool or statistical package was used to perform the GWAS.

The Mixed Model Analysis for Pedigree and population (MMA)⁴³ was used to perform all analysis and we updated the method section to clarify this.

References

1. Shen, H., Damcott, C.M., Rampersaud, E., Pollin, T.I., Horenstein, R.B., McArdle, P.F., Peyser, P.A., Bielak, L.F., Post, W.S., Chang, Y.P., et al. (2010). Familial defective apolipoprotein B-100 and increased low-density lipoprotein cholesterol and coronary artery calcification in the old order amish. *Archives of Internal Medicine* 170, 1850-1855.
2. Pollin, T.I., Damcott, C.M., Shen, H., Ott, S.H., Shelton, J., Horenstein, R.B., Post, W., McLenithan, J.C., Bielak, L.F., Peyser, P.A., et al. (2008). A null mutation in human APOC3 confers a favorable plasma lipid profile and apparent cardioprotection. *Science (New York, NY)* 322, 1702-1705.
3. Montasser, M.E., Hout, C.V.V., McFarland, R., Rosenberg, A., Callaway, M., Shen, B., Li, N., Daly, T.J., Howard, A.D., Lin, W., et al. (2019). Genetic and functional evidence relates a missense variant in B4GALT1 to lower LDL-C and fibrinogen. *BioRxiv preprint*.
4. Montasser, M.E., O'Hare, E.A., Wang, X., Howard, A.D., McFarland, R., Perry, J.A., Ryan, K.A., Rice, K., Jaquish, C.E., Shuldiner, A.R., et al. (2018). An APOO Pseudogene on Chromosome 5q Is Associated With Low-Density Lipoprotein Cholesterol Levels. *Circulation* 138, 1343-1355.
5. Perry, J.A. OASIS Resources, Video Library and Contact Information. <https://edn.som.umaryland.edu/OASIS/>.
6. Sherry, S.T., Ward, M.-H., Kholodov, M., Baker, J., Phan, L., Smigielski, E.M., and Sirotkin, K. (2001). dbSNP: the NCBI database of genetic variation. *Nucleic acids research* 29, 308-311.
7. Genome Aggregation Database (gnomAD). In. (
8. Carithers, L.J., and Moore, H.M. (2015). The genotype-tissue expression (GTEx) project. In. (Mary Ann Liebert, Inc. 140 Huguenot Street, 3rd Floor New Rochelle, NY 10801 USA.
9. Carvalho-Silva, D., Pierleoni, A., Pignatelli, M., Ong, C., Fumis, L., Karamanis, N., Carmona, M., Faulconbridge, A., Hercules, A., McAuley, E., et al. (2018). Open Targets Platform: new developments and updates two years on. *Nucleic Acids Research*, gky1133-gky1133.
10. Ionita-Laza, I., McCallum, K., Xu, B., and Buxbaum, J.D. (2016). A spectral approach integrating functional genomic annotations for coding and noncoding variants. *Nat Genet* 48, 214-220.
11. Boyle, A.P., Hong, E.L., Hariharan, M., Cheng, Y., Schaub, M.A., Kasowski, M., Karczewski, K.J., Park, J., Hitz, B.C., Weng, S., et al. (2012). Annotation of functional variation in personal genomes using RegulomeDB. *Genome research* 22, 1790-1797.
12. Zerbino, D.R., Wilder, S.P., Johnson, N., Juettemann, T., and Flicek, P.R. (2015). The ensembl regulatory build. *Genome Biol* 16, 56.
13. Varshney, A., Scott, L.J., Welch, R.P., Erdos, M.R., Chines, P.S., Narisu, N., Albanus, R.D., Orchard, P., Wolford, B.N., Kursawe, R., et al. (2017). Genetic regulatory signatures underlying islet gene expression and type 2 diabetes. *Proc Natl Acad Sci U S A* 114, 2301-2306.
14. Casper, J., Zweig, A.S., Villarreal, C., Tyner, C., Speir, M.L., Rosenbloom, K.R., Raney, B.J., Lee, C.M., Lee, B.T., and Karolchik, D. (2017). The UCSC genome browser database: 2018 update. *Nucleic acids research* 46, D762-D769.

15. Snyder, M.P., Gingeras, T.R., Moore, J.E., Weng, Z., Gerstein, M.B., Ren, B., Hardison, R.C., Stamatoyannopoulos, J.A., Graveley, B.R., Feingold, E.A., et al. (2020). Perspectives on ENCODE. *Nature* 583, 693-698.
16. Bycroft, C., Freeman, C., Petkova, D., Band, G., Elliott, L.T., Sharp, K., Motyer, A., Vukcevic, D., Delaneau, O., O'Connell, J., et al. (2018). The UK Biobank resource with deep phenotyping and genomic data. *Nature* 562, 203-209.
17. UKBiobank ICD PheWeb. 2019-06-30 <https://pheweb.org/UKB-SAIGE/>.
18. UK Biobank GWAS round 2 2019-03-30 <http://www.nealelab.is/uk-biobank>.
19. Pan-UK Biobank. 2020-11-30 <https://pan.ukbb.broadinstitute.org/>.
20. FinnGen Documentation of R4 release. 2020 2020-11-30 <https://finngen.gitbook.io/>.
21. Japanese Encyclopedida of Genetic associations by Riken <http://jenger.riken.jp/en/result>.
22. Tabassum, R., Rämö, J.T., Ripatti, P., Koskela, J.T., Kurki, M., Karjalainen, J., Palta, P., Hassan, S., Nunez-Fontarnau, J., Kiiskinen, T.T.J., et al. (2019). Genetic architecture of human plasma lipidome and its link to cardiovascular disease. *Nat Commun* 10, 4329.
23. Global Lipids Genetics, C., Willer, C.J., Schmidt, E.M., Sengupta, S., Peloso, G.M., Gustafsson, S., Kanoni, S., Ganna, A., Chen, J., Buchkovich, M.L., et al. (2013). Discovery and refinement of loci associated with lipid levels. *Nature genetics* 45, 1274-1283.
24. Liu, D.J., Peloso, G.M., Yu, H., Butterworth, A.S., Wang, X., Mahajan, A., Saleheen, D., Emdin, C., Alam, D., Alves, A.C., et al. (2017). Exome-wide association study of plasma lipids in >300,000 individuals. *Nature genetics* 49, 1758-1766.
25. Surakka, I., Horikoshi, M., Magi, R., Sarin, A.P., Mahajan, A., Lagou, V., Marullo, L., Ferreira, T., Miraglio, B., Timonen, S., et al. (2015). The impact of low-frequency and rare variants on lipid levels. *Nature genetics* 47, 589-597.
26. Teslovich, T.M., Musunuru, K., Smith, A.V., Edmondson, A.C., Stylianou, I.M., Koseki, M., Pirruccello, J.P., Ripatti, S., Chasman, D.I., Willer, C.J., et al. (2010). Biological, clinical and population relevance of 95 loci for blood lipids. *Nature* 466, 707-713.
27. Natarajan, P., Peloso, G.M., Zekavat, S.M., Montasser, M., Ganna, A., Chaffin, M., Khera, A.V., Zhou, W., Bloom, J.M., Engreitz, J.M., et al. (2018). Deep-coverage whole genome sequences and blood lipids among 16,324 individuals. *Nat Commun* 9, 3391.
28. Klarin, D., Damrauer, S.M., Cho, K., Sun, Y.V., Teslovich, T.M., Honerlaw, J., Gagnon, D.R., DuVall, S.L., Li, J., Peloso, G.M., et al. (2018). Genetics of blood lipids among ~300,000 multi-ethnic participants of the Million Veteran Program. *Nature genetics* 50, 1514-1523.
29. Montasser, M.E., Van Hout, C.V., Miloscio, L., Howard, A.D., Rosenberg, A., Callaway, M., Shen, B., Li, N., Locke, A.E., Verweij, N., et al. (2021). Genetic and functional evidence links a missense variant in. *Science* 374, 1221-1227.
30. Watanabe, K., Stringer, S., Frei, O., Umićević Mirkov, M., de Leeuw, C., Polderman, T.J.C., van der Sluis, S., Andreassen, O.A., Neale, B.M., and Posthuma, D. (2019). A global overview of pleiotropy and genetic architecture in complex traits. *Nat Genet* 51, 1339-1348.
31. Shikov, A.E., Skitchenko, R.K., Predeus, A.V., and Barbitoff, Y.A. (2020). Phenome-wide functional dissection of pleiotropic effects highlights key molecular pathways for human complex traits. *Sci Rep* 10, 1037.

32. Tachmazidou, I., Dedoussis, G., Southam, L., Farmaki, A.E., Ritchie, G.R., Xifara, D.K., Matchan, A., Hatzikotoulas, K., Rayner, N.W., Chen, Y., et al. (2013). A rare functional cardioprotective APOC3 variant has risen in frequency in distinct population isolates. *Nature communications* 4, 2872.
33. Tg, Hdl Working Group of the Exome Sequencing Project, N.H.L., Blood, I., Crosby, J., Peloso, G.M., Auer, P.L., Crosslin, D.R., Stitzel, N.O., Lange, L.A., Lu, Y., et al. (2014). Loss-of-function mutations in APOC3, triglycerides, and coronary disease. *The New England journal of medicine* 371, 22-31.
34. Saleheen, D., Natarajan, P., Armean, I.M., Zhao, W., Rasheed, A., Khetarpal, S.A., Won, H.H., Karczewski, K.J., O'Donnell-Luria, A.H., Samocha, K.E., et al. (2017). Human knockouts and phenotypic analysis in a cohort with a high rate of consanguinity. *Nature* 544, 235-239.
35. Reyes-Soffer, G., Sztalryd, C., Horenstein, R.B., Holleran, S., Matveyenko, A., Thomas, T., Nandakumar, R., Ngai, C., Karmally, W., Ginsberg, H.N., et al. (2019). Effects of APOC3 Heterozygous Deficiency on Plasma Lipid and Lipoprotein Metabolism. *Arteriosclerosis, Thrombosis, and Vascular Biology* 39, 63-72.
36. Paik, J., and Duggan, S. (2019). Volanesorsen: First Global Approval. *Drugs* 79, 1349-1354.
37. Vrablik, M., Tichý, L., Freiburger, T., Blaha, V., Satny, M., and Hubacek, J.A. (2020). Genetics of Familial Hypercholesterolemia: New Insights. *Front Genet* 11, 574474.
38. Beheshti, S., Madsen, C.M., Varbo, A., Benn, M., and Nordestgaard, B.G. (2018). Relationship of Familial Hypercholesterolemia and High Low-Density Lipoprotein Cholesterol to Ischemic Stroke: Copenhagen General Population Study. *Circulation* 138, 578-589.
39. Cajka, T., Smilowitz, J.T., and Fiehn, O. (2017). Validating Quantitative Untargeted Lipidomics Across Nine Liquid Chromatography-High-Resolution Mass Spectrometry Platforms. *Anal Chem* 89, 12360-12368.
40. Fan, S., Kind, T., Cajka, T., Hazen, S.L., Tang, W.H.W., Kaddurah-Daouk, R., Irvin, M.R., Arnett, D.K., Barupal, D.K., and Fiehn, O. (2019). Systematic Error Removal Using Random Forest for Normalizing Large-Scale Untargeted Lipidomics Data. *Anal Chem* 91, 3590-3596.
41. (2015). The Genotype-Tissue Expression (GTEx) pilot analysis: multitissue gene regulation in humans. *Science* 348, 648-660.
42. Vösa, U., Claringbould, A., Westra, H.J., Bonder, M.J., Deelen, P., Zeng, B., Kirsten, H., Saha, A., Kreuzhuber, R., Yazar, S., et al. (2021). Large-scale cis- and trans-eQTL analyses identify thousands of genetic loci and polygenic scores that regulate blood gene expression. *Nat Genet* 53, 1300-1310.
43. O'Connell, J.R. Mixed Model Analysis for Pedigree and population (MMAP) <https://github.com/MMAP> DOI:10.5281/zenodo.5033491.

REVIEWERS' COMMENTS:

Reviewer #1 (Remarks to the Author):

The authors have adequately addressed my major points. I do not have additional comments.

Reviewer #2 (Remarks to the Author):

The authors have addressed my comments appropriately, and have notably improved their manuscript.

I do not have any further suggestions.

Reviewer #3 (Remarks to the Author):

The authors have adequately addressed all the comments in the revised manuscript. I have no further comments.